# Learning to Solve Combinatorial Graph Partitioning Problems via Efficient Exploration

## Abstract

From logistics to the natural sciences, combinatorial optimisation on graphs underpins numerous real-world applications. Reinforcement learning (RL) has shown particular promise in this setting as it can adapt to specific problem structures and does not require pre-solved instances for these, often NP-hard, problems. However, state-of-the-art (SOTA) approaches typically suffer from severe scalability issues, primarily due to their reliance on expensive graph neural networks (GNNs) at each decision step. We introduce ECORD; a novel RL algorithm that alleviates this expense by restricting the GNN to a single pre-processing step, before entering a fast-acting exploratory phase directed by a recurrent unit. Experimentally, we demonstrate that ECORD achieves a new SOTA for RL algorithms on the Maximum Cut problem, whilst also providing orders of magnitude improvement in speed and scalability. Compared to the nearest competitor, ECORD reduces the optimality gap by up to $73\%$ on $500$ vertex graphs with a decreased wall-clock time. Moreover, ECORD retains strong performance when generalising to larger graphs with up to $10\,000$ vertices.

## 1 Introduction

Combinatorial optimisation (CO) problems seek to find the ordering, labelling, or subset of discrete elements that maximises some objective function. Despite this seemingly abstract mathematical formulation, CO arises at the heart of many practical applications, from logistics (Yanling et al., 2010) to protein folding (Perdomo-Ortiz et al., 2012) and fundamental science (Barahona, 1982). However, with these tasks often being NP-hard, solving CO problems becomes increasingly challenging for all but the simplest of systems. This combination of conceptual simplicity, computational complexity, and practical importance has made CO a canonical challenge, and motivated significant efforts into developing approximate and heuristic algorithms for these tasks. Whilst approximate methods can offer guarantees on the solution quality, in practice they frequently lack sufficiently strong bounds and have limited scalability (Williamson & Shmoys, 2011). By contrast, well-designed heuristics offer no such guarantees but can prove highly efficient (Halim & Ismail, 2019).

As a result, recent years have seen a surge in the application of automated learning methods that parameterise CO heuristics with deep neural networks (Bengio et al., 2021). In particular, reinforcement learning (RL) has become a popular paradigm, as it can facilitate the discovery of novel heuristics without the need for labelled data. Moreover, many CO problems are naturally formulated as Markov decision processes (MDPs) on graphs, where vertices correspond to discrete variables and edges denote their interaction or dependence. Accordingly, graph neural networks (GNNs) have become the de-facto function approximator of choice as they reflect the underlying structure of the problem whilst seamlessly handling variable problem sizes and irregular topological structures.

However, despite the demonstrated success of RL-GNN approaches, scalability remains an outstanding challenge. Running a GNN for every decision results in impractical computational overheads for anything beyond small- to medium-sized problems. This is exacerbated by the fact that directly predicting the solution to an NP-hard problem is typically unrealistic. As such, leading approaches often utilise stochastic exploration or structured search to generate multiple candidate solutions (Chen & Tian, 2019; Joshi et al., 2019; Gupta et al., 2020; Barrett et al., 2020; Bresson & Laurent, 2021) – which ultimately requires longer solving times and an increased computational burden.

The notion of exploratory combinatorial optimisation – reframing the task from predicting a single solution to exploring the solution space using reversible single-vertex actions (Barrett et al., 2020) – also hints at a mitigation to the scalability issue it highlights. Intuitively, when any decisions can be reversed, the quality of any single decision is less critical, so long as overall improvements are made over the course of many such actions. In this work, we leverage this notion to introduce a new algorithm, ECORD (Exploratory Combinatorial Optimisation with Rapid Decoding), that combines a single GNN preprocessing step with fast action-decoding that replaces further geometric inference with simple per-vertex observations and a learnt representation of the ongoing optimisation trajectory. The result is a theoretical and demonstrated speed-up over expensive GNN action-decoding with the action-selection time of ECORD being independent of the graph topology and, in practice, near constant regardless of graph size.

Experimentally, we consider the Maximum Cut (Max-Cut) problem, a canonical CO problem chosen because of its generality (11 of the 21 NP-complete problems presented by Karp (1972) can be reduced to Max-Cut, including graph coloring, clique cover, knapsack and Steiner Tree) and the fact that it presents a challenging problem for scalable CO as the optimal solution requires every vertex to be correctly labelled (rather than simply a subset). This combination of wide-ranging applicability and intractability has motivated significant commercial and research efforts into Max-Cut solvers, from bespoke hardware based on classical (Goto et al., 2019) and quantum annealing (Yamamoto et al., 2017; Djidjev et al., 2018) to hand-crafted (Goemans & Williamson, 1995; Benlic & Hao, 2013) or learnt (Barrett et al., 2020) heuristic algorithms.

ECORD is found to equal or surpass the performance of expensive SOTA RL-GNN methods on graphs with up to $500$ vertices (where all methods are computationally feasible), even when compared on number of actions instead of wall-clock time. Moreover, the low computational overhead of ECORD is seen to provide orders of magnitude improvements in speed and scalability, with strong performance, and a nearly $300\times$ increase in throughput when compared to conventional RL-GNN methods, demonstrated on graphs with up to $10\,000$ vertices.

## 2 RELATED WORK

The application of neural networks to graph-based CO problems dates back to Hopfield & Tank (1985) who considered the Travelling Salesman Problem (TSP). However RL techniques were not applied until a decade later in the work of Zhang & Dietterich (1995) on the NP-hard job-shop problem. More recently, the advancement of deep learning and RL has triggered a resurgence in the ML community's interest in developing CO solvers, with multiple reviews providing detailed taxonomies (Bengio et al., 2021; Mazyavkina et al., 2020; Vesselinova et al., 2020).

**Learning to solve non-Euclidean CO**. This resurgence began by considering Euclidean approaches that did not reflect the underlying graph structure of the problems. In this context, Bello et al. (2016) used RL to train pointer networks (PNs), which treat the discrete variables of the CO problem as a sequence of inputs to a recurrent unit (Vinyals et al., 2015) to solve TSP. By avoiding the need for labelled data sets, they were able to scale beyond the $40$ vertex limit of Vinyals et al. (2015), inferring on 20 to 100 vertex graphs. Gu & Yang (2020) further scaled this approach to instances of up to 300, using a hybrid supervised-reinforcement learning framework that combined PNs and A3C (Mnih et al., 2016). However, although PNs can handle graphs of different sizes (with the help of manual input/output engineering, such as zero padding), these Euclidean approaches fail to capture the topological structures and intricate relationships contained within graphs, and typically require a large number of training instances in order to generalise.

This issue was addressed by Dai et al. (2017), who trained a Structure-to-Vector (S2V) GNN with DQN to solve TSP and Max-Cut. The resulting algorithm, S2V-DQN, generalised to graphs with different size and structure to the training set and achieved excellent performance across a range of problems without the need for manual algorithmic design, demonstrating the value in exploiting underlying graph structure.

**Advances in optimality**. Various works since Dai et al. (2017) have sought to harness GNN embeddings to improve solution quality. Abe et al. (2019) combined a GNN with a Monte Carlo tree search approach to learn a high-quality constructive heuristic. Ultimately, they demonstrated a greater ability to generalise to more graph types than S2V-DQN on Max-Cut, but their method could only scale

to 100 vertex Erdős-Rényi and Barabasi-Albert graphs. Li et al. (2018) combined a graph convolution network (GCN) with guided tree search to synthesise a diverse set of solutions and thereby more fully explore the space of possible solutions. However, they used supervised learning and so required labelled data which limits scalability, and they did not consider Max-Cut. Barrett et al. (2020) proposed ECO-DQN, the SOTA RL algorithm for Max-Cut, which reframed the role of the RL agent to looking to improve on any given solution, rather than directly predict the optimal solution. The key insight is that exploration *at test time* is beneficial, since individual sub-optimal decisions (which are to be expected for NP-hard problems) need not matter so long as the final solution is of high quality. However, ECO-DQN utilises an expensive GNN at each decision step and extends the overall number of decisions taken to be theoretically limitless, thereby restricting its scalability to be even worse than that of S2V-DQN. ECORD remedies this by using an initial GNN embedding followed by a recurrent unit to balance the richness provided by graph networks with fast-action selection.

**Advances in scalability**. Manchanda et al. (2020) furthered the work of Dai et al. (2017) by first training an embedding GCN in a supervised manner, and then training a Q-network with RL to predict per-vertex action values. By using the GCN embeddings to prune nodes unlikely to be in the solution set, their method provided significantly more scalability than S2V-DQN on the the Maximum Vertex Cover problem. However, it was not applicable to problems whose nodes cannot be pruned, which precludes it from solving some of the most fundamental CO problems such as Max-Cut. Drori et al. (2020) took a different approach, proposing a general RL-GNN framework that uses a graph attention network to create a dense embedding of the input graph followed by a recurrent attention mechanism for action selection. They achieved impressive scalability, reaching instance sizes of 1000 vertices on the minimum spanning tree problem and TSP. However, unlike ECORD, the Drori et al. (2020) framework restricts the decoding stage to condition only on the previously selected action, and considers only node ordering problems in a non-exploratory setting.

## 3 METHODS

### 3.1 BACKGROUND

**Max-Cut problem**. The Max-Cut problem seeks a binary labelling of all nodes in a graph, such that the number (or cumulative weight) of edges connecting nodes of opposite labels is maximised. Concretely, for a graph, $G(V, E)$, with vertices $V$ and edges $E$, we wish to find a subset of vertices, $S \subset V$ that maximises the 'cut-value', $C(S, G) = \sum_{i \in S, j \in V \setminus S} w(e_{ij})$, where $w(e_{ij})$ is the weight of an edge $e_{ij} \in E$.

**Q-learning**. We formulate the optimisation as a Markov Decision Process (MDP) defined by a tuple $\{\mathcal{S}, \mathcal{A}, \mathcal{T}, \mathcal{R}, \gamma\}$, where $\mathcal{S}$ and $\mathcal{A}$ are the state and action spaces, respectively, $\mathcal{T} : \mathcal{S} \times \mathcal{A} \times \mathcal{S} \rightarrow [0, 1]$ is the transition function, $\mathcal{R} : \mathcal{S} \rightarrow \mathbb{R}$ the reward function and $\gamma \in [0, 1]$ the discount factor. Q-learning methods seek to directly learn the Q-value function mapping state-action pairs, $(s \in \mathcal{S}, a \in A)$, to the expected discounted sum of immediate and future rewards when following a policy $\pi : \mathcal{S} \rightarrow \mathcal{A}$, $Q^{\pi}(s, a) = \mathbb{E}_{\pi}[\sum_{t'=t+1}^{\infty} \gamma^{t'-1} r(s^{(t')})|s^{(t)}{=}s, a^{(t)}{=}a]$. Conventional DQN (Mnih et al., 2015) optimises $Q_{\theta}$ by minimising the mean-squared-error between the network predictions and a bootstrapped estimate of the Q-value. By definition, an optimal policy maximises the true Q-value of every selected action, therefore, after training, an approximation of an optimal policy is obtained by acting greedily with respect to the learnt value function, $\pi_{\theta}(s) = \arg\max_{a'} Q_{\theta}(s, a')$.

**Munchausen DQN**. Munchausen DQN (M-DQN) (Vieillard et al., 2020) makes two fundamental adjustments to conventional DQN. Firstly, the Q-values are considered to define a stochastic policy with action probabilities given by $\pi_{\theta}(\cdot|s) = \text{softmax}(\frac{Q_{\theta}(s, \cdot)}{\tau})$, where $\tau$ is a temperature parameter. Secondly, M-DQN adds the log-probability of the selected action to the reward at each step. All together, the regression target for the Q-function is modified to

$$Q_{\text{m-dqn}}(s^{(t)}, a^{(t)}, s^{(t+1)}) = r(s^{(t+1)}) + \alpha\tau \ln \pi_{\theta}(a^{(t)}|s^{(t)}) +$$
$$\gamma \mathbb{E}_{a' \sim \pi(\cdot|s^{(t+1)})}[Q_{\theta}(s^{(t+1)}, a') - \tau \ln \pi_{\theta}(a'|s^{(t+1)})], \quad (1)$$

where $\alpha$ scales the additional log-policy contribution to the reward. Note that as $\alpha \rightarrow 0$ and $\tau \rightarrow 0$ we recover the standard DQN regression target.

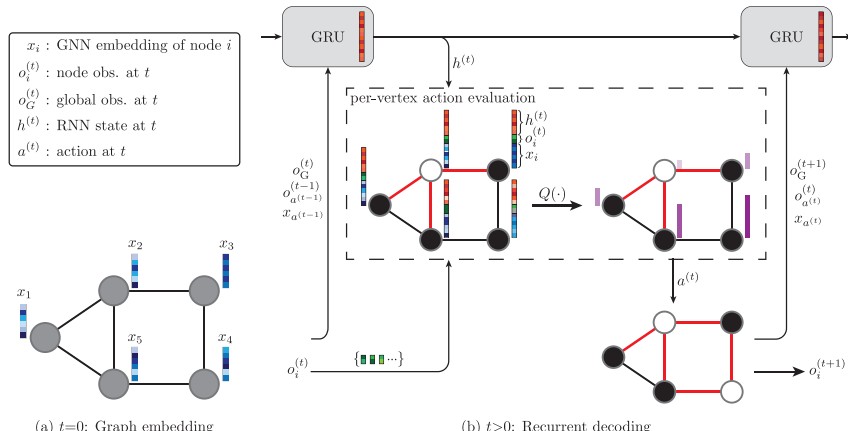

(a) $t$=0: Graph embedding      (b) $t$>0: Recurrent decoding

Figure 1: Diagram of the proposed ECORD architecture applied to solving a Max-Cut instance. (a) A GNN generates initial embeddings for each vertex at $t = 0$. (b) At each step in the episode, a recurrent unit takes as input the initial graph embedding, the node-level observation from the previous time step, and the global observation from the current time step, and updates the hidden state vector. To select an action (node to flip), the hidden state is concatenated with the node-level observation at the current time step and passed to a Q-network.

## 3.2 ECORD

**MDP formulation**. The state, $s^{(t)} \equiv (G(V,E), S) \in \mathcal{S}$, at a given time-step corresponds to the graph and a binary labelling of all vertices. An action selects a single vertex and 'flips' its label to the opposite state, adding or removing it from the solution subset. Rewards correspond to the (normalised and clipped) increase in best cut-value, $r(s^{(t)}) = \max((C(s^{(t)}) - C(s^*)))/|V|, 0)$, where $C(s^*)$ is the the best solution found in the episode thus far. This choice of reward structure motivates the agent to maximise the cut-value at any step within the episode, without discouraging it from searching through unseen solutions. Naturally, the transition function is known and fully deterministic for our mathematically defined optimisation problem.

Concretely, we represent the state, $s^{(t)}$, with per-vertex observations, $o_i^{(t)} \in \mathbb{R}^3$ for $i \in V$, and a global observation of the current state, $o_G^{(t)} \in \mathbb{R}^2$. For each vertex, we provide: (i) the current label, (ii) the immediate change in cut-value if the vertex is flipped (which we refer to as a 'peek') and (iii) the number of steps since the node was last flipped. Our global observations are the (normalised) difference in cut-value between the current and best observed state, and the maximum peek of any vertex in the current state.

Importantly, these features are readily available without introducing any significant computational burden. Indeed, even the peeks – one-step look-aheads for each action – can be calculated for all vertices once at the beginning of each episode, after which they can be efficiently updated quickly at each step for only the flipped vertex and it's immediate neighbours (details can be found appendix A.3). Besides this being a cheap operation, this approach is also the most efficient way to track the cut-value over the course of multiple vertex flips, and thus is a necessary calculation to evaluate each environmental state regardless of whether the peeks form part of the observation.

**Architecture**. ECORD splits an optimisation into two stages (see figure 1 for an overview of the entire architecture). (i) A GNN prepares per-vertex embeddings conditioned on only the geometric structure (i.e. weighted adjacency matrix) of the graph itself. (ii) Starting from a random labelling of the vertices, ECORD sequentially flips vertices between subsets, with each action conditioned on the static GNN embeddings, the previous trajectory steps (through the hidden state of an RNN) and simple observations of the current state (described above). Full details of the architecture can be found in appendix A.1, with this section providing a higher level summary.

To generate the per-vertex embeddings, we use a gated graph convolution network (Li et al., 2015). The final embeddings after $L$ rounds of message-passing are linearly projected to per-vertex embeddings, $x_i = W_{\mathrm{p}}x_i^{(L)}$ where $i$ denotes the vertex, that remain fixed for the remainder of the episode.

To select an action at time $t$, we first combine the GNN outputs with the observation to obtain per-vertex embeddings of the form $v_i^{(t)} = [x_i, W_{\mathrm{o}}o_i^{(t)}]$, where square brackets denote concatenation. This local information is then used in conjunction with the RNN hidden state, $h^{(t)}$, to predict the Q-value for flipping vertex $i$. The Q-network itself uses a duelling architecture (Wang et al., 2016), where separate MLPs estimate the value of the state, $V(\cdot)$, and advantage for each action, $A(\cdot)$,

$$Q(v_i^{(t)}, h^{(t)}) = V(h^{(t)}) + A(v_i^{(t)}, h^{(t)}) = \mathrm{MLP_V}(h^{(t)}) + \mathrm{MLP_A}([v_i^{(t)}, W_{\mathrm{h}}h^{(t)}]). \tag{2}$$

$h^{(t)}$ is shared across each vertex when calculating the advantage function as our reward function depends on the current best score within the trajectory's history, a global feature contained in the state-level embedding. However, before concatenating them together, we project the high dimensional (1024 in our case) hidden state to match the lower dimensional (32) per-vertex embeddings.

Finally, we update the RNN using the embedding of the selected action, $v_*^{(t)}$, and our global observation after taking it, $o_{\mathrm{G}}^{(t+1)}$,

$$h^{(t+1)} = \mathrm{GRU}(h^{(t)}, m^{(t+1)}), \quad \text{where} \quad m^{(t+1)} = \mathrm{MLP_m}([v_*^{(t)}, o_{\mathrm{G}}^{(t+1)}]). \tag{3}$$

**Training**. ECORD is trained using M-DQN. As the Q-values are conditioned upon the internal state of an RNN, see eq. (2), we train using truncated backpropagation through time (BPTT). Concretely, when calculating the Q-values at time $t$ during training, we reset the environment and the internal state of the RNN to their state at time $t - k_{\mathrm{BPTT}}$ and replay the trajectory to time $t$. In doing so, we only have to backpropagate through the previous $k_{\mathrm{BPTT}}$ time steps when minimising the loss.

The intuition behind M-DQN's modifications are (i) the entropy of the policy is jointly optimised with the returns, in the spirit of maximum entropy RL (Haarnoja et al., 2018), and (ii) the agent is rewarded for being more confident about the correct action. The second point is based on the assumption that an optimal policy is deterministic, since it will always take the action with maximum Q-value. One could observe that as ECORD's action space is large (equal to the number of vertices in the graph, $|V|$) the structured exploration of M-DQN's stochastic policy may allow for more meaningful trajectories than the standard epsilon-greedy approaches used in DQN. Moreover, the underlying postulate of M-DQN, that despite the stochastic exploration policy, the true optimal policy is deterministic, aligns exactly with our problem setting.

## 4 EXPERIMENTS

**Baselines**. We compare ECORD to the previous two SOTA RL-GNN algorithms for Max-Cut, S2V-DQN (Khalil et al., 2017) and ECO-DQN (Barrett et al., 2020). Our implementation of ECORD contains several speed improvements in comparison to the publicly available implementation of ECO-DQN (e.g. parallelised optimisation trajectories, compiled calculations, sparse matrix operations). To ensure the fairest possible comparison, when directly comparing performance independent of speed (section 4.1), we use the public implementation, however when comparing scalability (section 4.2 and 4.3) we use a re-implementation of ECO-DQN within the same codebase as ECORD.

Additionally, a simple heuristic, Max-Cut Approx (MCA), that greedily selects actions that maximise the immediate increase in cut value is also considered. Besides from providing surprisingly strong performance, the choice of MCA baselines is motivated by the observation that the one-step look-ahead 'peek' features make learning a greedy policy straightforward. Moreover, whilst MCA terminates once a locally optimal solution is found (i.e. one from which no single action can further increase the cut value), a network learning to only approximate a greedy policy may fortuitously escape these solutions and ultimately obtain better results. To address this concern, we introduce a simple extension of (and significant improvement over) MCA called MCA-soft, where actions are selected by a soft-greedy policy with the temperature tuned to maximise performance on the target dataset (see appendix A.4 for details).

**Datasets**. We consider graph datasets for which the optimal (or best known) solutions are publicly available. The dataset published by Barrett et al. (2020) consists of Erdős-Rényi (Erdős & Rényi,

1960) and Barabasi-Albert (Albert & Barabási, 2002) graphs (ER and BA, respectively) with edge weights $w(e_{ij}) \in \{0, \pm 1\}$ and up to 500 vertices. Each graph type and size is represented by 150 random instances, with 50 used for model selection and results reported on the remaining 100 at test time. We refer to these distributions as ER40/BA40 to ER500/BA500.

To test on larger graphs we use the GSet (Benlic & Hao, 2013), a well-studied dataset containing multiple distribution, from which we focus on random (ER) graphs with binary edge weights and 800 to 10 000 vertices. To aide model selection and parameter tuning, we generate 10 additional graphs for each distribution in the GSet. We refer to these validation sets as ER800 to ER10000.

**Metrics**. Our analysis considers the raw performance, wall-clock speed, and memory usage of algorithms. Following prior work, we use the approximation ratio, given by $AR(s^*) = C(s^*)/C_{\text{opt}}$ where $C_{\text{opt}}$ is the best known cut value, as a metric of solution quality. All experiments were performed on the same system with an Nvidia GeForce RTX 2080 Ti 11 GB GPU and 80 processors (Intel(R) Xeon(R) Gold 6248 CPU @ 2.50 GHz).

**Reproducability**. All datasets considered in this work are either available or linked in the supporting code at `https://github.com/[MASKED-FOR-BLIND-REVIEW]`, along with source code and scripts to reproduce the reported results.

## 4.1 COMPARISON TO SOTA RL METHODS

Table 1: Agent scores trained on ER40 (best in **bold**). Error bars denote $68\,\%$ confidence intervals. RL baseline scores are taken directly from Barrett et al. (2020) for fair comparison. Results are averaged across 5 random seeds for the RL baselines and 3 random seeds for ECORD.

|  | Heuristics | | RL baselines | | |
|---|---|---|---|---|---|
|  | MCA | MCA-soft | S2V-DQN | ECO-DQN | ECORD |
| ER40 | $0.997^{+0.001}_{-0.010}$ | $\mathbf{1.000}^{+0.000}_{-0.000}$ | $0.980^{+0.014}_{-0.023}$ | $\mathbf{1.000}^{+0.000}_{-0.000}$ | $\mathbf{1.000}^{+0.000}_{-0.000}$ |
| ER60 | $0.994^{+0.003}_{-0.013}$ | $0.999^{+0.001}_{-0.004}$ | $0.973^{+0.021}_{-0.024}$ | $\mathbf{1.000}^{+0.000}_{-0.000}$ | $\mathbf{1.000}^{+0.000}_{-0.000}$ |
| ER100 | $0.977^{+0.017}_{-0.017}$ | $0.993^{+0.007}_{-0.008}$ | $0.961^{+0.029}_{-0.028}$ | $\mathbf{1.000}^{+0.000}_{-0.001}$ | $\mathbf{1.000}^{+0.000}_{-0.001}$ |
| ER200 | $0.959^{+0.014}_{-0.014}$ | $0.973^{+0.012}_{-0.012}$ | $0.951^{+0.020}_{-0.020}$ | $0.999^{+0.000}_{-0.002}$ | $\mathbf{1.000}^{+0.000}_{-0.001}$ |
| ER500 | $0.941^{+0.012}_{-0.012}$ | $0.958^{+0.009}_{-0.009}$ | $0.921^{+0.019}_{-0.019}$ | $0.985^{+0.006}_{-0.006}$ | $\mathbf{0.996}^{+0.004}_{-0.004}$ |
| BA40 | $0.999^{+0.000}_{-0.006}$ | $\mathbf{1.000}^{+0.000}_{-0.000}$ | $0.967^{+0.023}_{-0.040}$ | $\mathbf{1.000}^{+0.000}_{-0.000}$ | $\mathbf{1.000}^{+0.000}_{-0.000}$ |
| BA60 | $0.989^{+0.007}_{-0.018}$ | $0.997^{+0.003}_{-0.008}$ | $0.968^{+0.022}_{-0.036}$ | $\mathbf{1.000}^{+0.000}_{-0.000}$ | $\mathbf{1.000}^{+0.000}_{-0.000}$ |
| BA100 | $0.965^{+0.020}_{-0.020}$ | $0.984^{+0.012}_{-0.013}$ | $0.940^{+0.032}_{-0.033}$ | $\mathbf{1.000}^{+0.000}_{-0.000}$ | $\mathbf{1.000}^{+0.000}_{-0.000}$ |
| BA200 | $0.911^{+0.033}_{-0.037}$ | $0.929^{+0.034}_{-0.034}$ | $0.865^{+0.058}_{-0.061}$ | $0.978^{+0.014}_{-0.033}$ | $\mathbf{0.983}^{+0.017}_{-0.033}$ |
| BA500 | $0.889^{+0.015}_{-0.015}$ | $0.899^{+0.014}_{-0.014}$ | $0.744^{+0.052}_{-0.052}$ | $\mathbf{0.967}^{+0.014}_{-0.015}$ | $0.963^{+0.012}_{-0.012}$ |

**Methods**. Our first set of experiments compares ECORD to both heuristic node-flipping, and SOTA RL, baselines. To facilitate a fair and direct comparison, we use exactly the models, datasets, and results published by Barrett et al. (2020) for S2V-DQN and ECO-DQN. All RL algorithms are trained on 40 vertex ER graphs and evaluated on both ER and BA graphs with up to 500 vertices. S2V-DQN is deterministic and incrementally constructs the solution set one vertex at a time, therefore each optimisation trajectory consists of $|V|$ sequential actions. MCA-soft, ECO-DQN and ECORD allow any vertex to be 'flipped' at each step, and therefore can in principle run indefinitely on a target graph. The baselines presented are chosen as they consider the same sequential node-flipping paradigm as ECORD, however their exits multiple algorithms for solving Max-Cut that do not fit this paradigm – notably simulated annealing (SA) (Tiunov et al., 2019; Leleu et al., 2019), semidefinite programming (SDP) (Goemans & Williamson, 1995) and mixed integer programming (CPLEX, ILOG, 2007). We provide additional results using SOTA or commercial algorithms spanning these paradigms in appendix A.5, where ECORD is found to outperform SDP and MIP and either beat or be competitive with SOTA SA methodologies.

In practice, we use ECORD and the MCA heuristics with ECO-DQN's default settings; 50 optimisation trajectories per graph, each starting from a random node labelling, acting greedily with respect to the learnt Q-values, and terminating after $2|V|$ sequential actions. We note that this disadvantages ECORD as (i) despite learning a stochastic policy, ECORD acts deterministically at test time, and (ii) ECORD's significant speed and memory advantages over ECO-DQN are not accounted for.

**Results**. Despite the disadvantages described above, ECORD either outperforms or essentially matches ECO-DQN on all tests (see table 1), whilst significantly improving over all other baselines

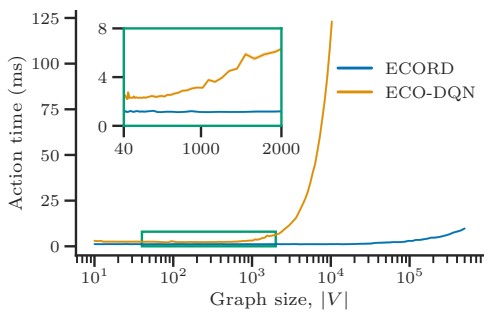

(a) Inference speed for single action.

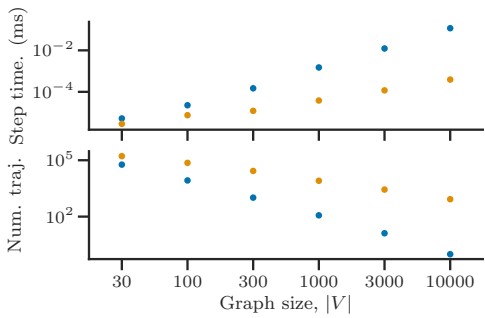

(b) Practical throughput for batched trajectories.

Figure 2: ECORD (blue) vs. ECO-DQN (orange) scaling performance. (a) Action selection time when running a single trajectory. (b) Step time (action selection and environment update time) when running the maximum number of tradjectories that fit on the GPU. ECORD's relative throughput at $|V| = \{30, 100, 300, 1000, 3000, 10\,000\}$ is $\{1.8, 3.0, 12.2, 39.3, 104.5, 298.9\}\times$ ECO-DQN's (beyond $|V|\approx10\,000$, ECO-DQN no longer fits on the GPU).

and generalising to unseen graph topologies and sizes. This seems surprising when we consider that, in contrast to ECO-DQN, ECORD does not directly condition per-node decisions on the state of other nodes. However, ECORD does have access to the optimisation trajectory via the RNN hidden state, suggesting that in exploratory settings, the temporal structure (in addition to the geometric structure of the graph) is highly informative. This experiment was repeated for a model trained instead on BA40 with the results (provided in appendix A.5) being qualitatively the same.

## 4.2 COMPUTATIONAL COMPLEXITY AND PRACTICAL SCALING PERFORMANCE

Whilst ECORD has already been shown to match or surpass SOTA RL baselines when ignoring computational cost, this is not a sufficient metric of algorithmic utility. In this section, we consider the theoretical complexity and practical scaling cost of ECORD.

**Theoretical complexity**. The runtime complexity of any optimisation trajectory scales linearly with the number of actions. In CO problems where every vertex must be correctly labelled, such as Max-Cut, episode length scales at best linearly with the number of nodes. A typical approach, which encapsulates both S2V-DQN and ECO-DQN, for applying RL to CO on graphs parameterises the policy or value function with a GNN. The precise complexity of a GNN depends on the chosen architecture and graph topology, however, typically the per-layer performance scales linearly with the number of edges in the graph, as this is the number of unique communication channels along which information must be passed. In practice, this results in a (worst case and typical) polynomial scaling of $\mathcal{O}(|V|^2)$ per-action and $\mathcal{O}(|V|^3)$ per optimisation, which makes even modest sized graphs with the order of hundreds of nodes very computationally expensive in practice.

In contrast, ECORD only runs the GNN once, regardless of the graph size or episode length, and then selects actions without any additional message passing between nodes. As a result, per-action computational cost scales linearly as $\mathcal{O}(|V|)$, regardless of the graph topology, and the entire optimisation scales as $\mathcal{O}(|V|^2)$. Moreover, action selection in ECORD also has a far smaller memory footprint than using a GNN over the entire graph, and each vertex can be processed in parallel up to the limits of hardware. Therefore, in practice, we typically obtain a constant, $\mathcal{O}(1)$, scaling of the per-action computational cost and, as the single graph network pass is typically negligible compared to the long exploratory phase, complete the entire optimisation in $\mathcal{O}(|V|)$.

**Practical performance**. Figure 2 demonstrates the practical performance realised from these theoretical improvements. In 2a, the action time of ECORD and ECO-DQN are compared on graphs with up to $500\,\text{k}$ vertices. Whilst ECORD is always faster, both take near constant time for small graphs where batched inference across all nodes is still efficient. However, even when ECO-DQN begins to increase in cost ($|V|\approx500$) and eventually fills the entire GPU memory ($|V|\approx10\,\text{k}$), ECORD retains a fixed low-cost inference which only appreciably begins to increase for large graphs ($|V|>100\,\text{k}$).

Moreover, the reduced memory footprint of ECORD also allows for more (randomly initialised) optimisation trajectories to be run in parallel. Figure 2b contrasts ECORD and ECO-DQN by running as many parallel trajectories as possible for a single graph with up to $|V|=10\,\mathrm{k}$, and plotting the effective step time (action selection plus environmental step) per trajectory. The practical increase in throughput of ECORD compared to ECO-DQN increases with graph size from a modest $1.8\times$ with $|V|=30$ to $100\times$ for $|V|=3\,\mathrm{k}$ and $298.9\times$ for $|V|=100\,\mathrm{k}$.

### 4.3 SCALING TO LARGE GRAPHS

**Methods**. ECORD is trained on graphs with binary edge weights, $w(e_{ij}) \in \{0,1\}$ and $|V| = 500$. The optimal parameters are selected according to the performance of the greedy policy on the generated ER10000 set. For the computational reasons discussed previously, ECO-DQN was trained on ER200 graphs with binary edge weights, and used ER500 graphs for model selection. The temperature of MCA-soft was tuned independently on each graph, as detailed in appendix A.4.

**Stochastic exploration**. Despite ECORD using a stochastic soft-greedy policy during training, at test time we previously used a deterministic policy (which, ultimately, still provided near optimal performance on small and intermediate graphs). To investigate the performance of a soft-greedy policy, we evaluated ECORD on the large-graph datasets ER5000, ER7000 and ER10000 – using 20 trajectories of $4|V|$ steps for all graphs in the test sets – across a range of temperatures.

The results are summarised in figure 3. The key points are that a non-zero temperature is optimal for larger graphs, and that the optimal temperature decreases with graph size. The interpretation is that even an exploratory agent such as ECORD can still eventually get stuck in closed trajectories when acting deterministically. Therefore, introducing some stochasticity allows ECORD to escape these regions of the solution space and ultimately continue to search for improved configurations. However, this must be balanced with the need for a sufficiently deterministic policy that a good solution is found with high probability. Intuitively, longer sequences of actions are required to reach a good configuration for larger graphs, which aligns with the observed dependence of the optimal temperature with graph size.

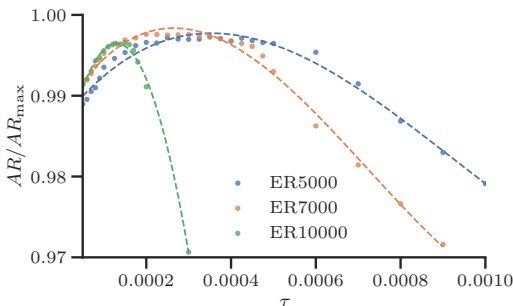

Figure 3: Effect of temperature on ECORD's performance for $|V| = \{5000, 7000, 10\,000\}$ graphs. The dashed lines are simply to guide the eye.

**GSet results**. We validate ECORD's performance on large graphs by testing it on all GSet graphs with random binary edges, as summarised in table 2. For each graph we run 20 parallel optimisation trajectories for a maximum of $180\,\mathrm{s}$. For G5000, G7000 and G10000 we use tuned temperatures, $\tau$, of 3.5e−4, 2e−4 and 1e−4 respectively, and set $\tau=5\mathrm{e}{-4}$ for all other graphs.

ECORD shows consistently strong performance, matching or beating ECO-DQN and MCA-soft on every instance by a margin that increases with graph size. The speed of ECORD is emphasised by the fact that it obtains approximation ratios of above 0.99 on all graphs with $|V| \leq 2\,\mathrm{k}$ in under $10\,\mathrm{s}$. Whilst in principle it is possible to reach the optimal solution in $|V|$ actions, on larger graphs the obtained cut values consistently increase with longer exploration (for reference, ECORD takes $>28\,\mathrm{k}$ actions in $180\,\mathrm{s}$ when $|V|=10\,\mathrm{k}$). This clearly demonstrates that ECORD learns to search through the solution space without getting stuck, rather than generating a single mode of solutions.

A natural final question is whether the slight decrease in performance on the largest graphs is due to an insufficient time budget, or the learnt policy being less suitable for these problems. To test this, ECORD was allowed to continue optimising G70 for $1\,\mathrm{h}$. Whilst the obtained solution improved to have an approximation ratio of $0.978$ (obtained in only $233\,\mathrm{s}$), the optimal solution was not found. Ultimately, whilst previous RL-GNN methods were limited by the scalability of GNNs, now we find ourselves limited by the ability of the agent to reason about larger systems – opening the door to, and defining a challenge for, future research.

Table 2: ECORD, ECO-DQN and MCA-soft performance on the GSet graphs given a time budget of 10, 30, and 180 s (best in **bold**).

| Graph | $|V|$ | MCA-soft | | | ECO-DQN | | | ECORD | | |
|---|---|---|---|---|---|---|---|---|---|---|
| | | 10 s | 30 s | 180 s | 10 s | 30 s | 180 s | 10 s | 30 s | 180 s |
| G1 | 800 | 0.994 | 0.999 | 0.999 | 0.993 | 0.993 | 0.998 | 0.998 | **1.000** | — |
| G2 | 800 | 0.995 | 0.996 | 0.998 | 0.986 | 0.995 | 0.996 | 0.998 | 0.998 | **1.000** |
| G3 | 800 | 0.996 | 0.999 | 0.999 | 0.992 | 0.993 | 0.996 | 0.999 | **1.000** | — |
| G4 | 800 | 0.996 | 0.997 | 0.999 | 0.985 | 0.992 | 0.998 | 0.999 | **1.000** | — |
| G5 | 800 | 0.996 | 0.997 | 0.997 | 0.993 | 0.994 | 0.998 | **1.000** | — | — |
| G43 | 1000 | 0.990 | 0.997 | **1.000** | 0.985 | 0.992 | 0.998 | 0.996 | 0.997 | **1.000** |
| G44 | 1000 | 0.992 | 0.995 | 0.999 | 0.987 | 0.994 | 0.996 | 0.999 | 0.999 | **1.000** |
| G45 | 1000 | 0.995 | 0.998 | 0.998 | 0.988 | 0.992 | 0.996 | 0.998 | 0.999 | **0.999** |
| G46 | 1000 | 0.992 | 0.994 | 0.998 | 0.990 | 0.992 | 0.997 | 0.998 | 0.999 | **1.000** |
| G47 | 1000 | 0.996 | 0.997 | **0.998** | 0.988 | 0.993 | 0.994 | 0.997 | 0.998 | **0.998** |
| G22 | 2000 | 0.985 | 0.990 | 0.995 | 0.966 | 0.981 | 0.991 | 0.990 | 0.995 | **0.997** |
| G23 | 2000 | 0.985 | 0.992 | 0.996 | 0.967 | 0.983 | 0.989 | 0.991 | 0.995 | **0.997** |
| G24 | 2000 | 0.986 | 0.991 | **0.997** | 0.967 | 0.983 | 0.989 | 0.992 | 0.994 | **0.997** |
| G25 | 2000 | 0.985 | 0.993 | 0.997 | 0.966 | 0.985 | 0.990 | 0.991 | 0.996 | **0.998** |
| G26 | 2000 | 0.987 | 0.993 | 0.995 | 0.963 | 0.982 | 0.984 | 0.991 | 0.996 | **0.999** |
| G55 | 5000 | 0.952 | 0.968 | 0.969 | 0.832 | 0.873 | 0.939 | 0.947 | 0.969 | **0.985** |
| G60 | 7000 | 0.934 | 0.963 | 0.968 | 0.764 | 0.847 | 0.930 | 0.916 | 0.960 | **0.981** |
| G70 | 10 000 | 0.865 | 0.934 | 0.957 | 0.677 | 0.818 | 0.898 | 0.837 | 0.944 | **0.972** |

**Ablations**. ECORD's key components are: (i) the use of a single (GNN) encoding step to embed the problem structure, (ii) rapid decoding steps where per-vertex actions are conditioned on only local observations and an (RNN) leant embedding of the optimisation trajectory, and (iii) an exploratory CO setting where actions can be reversed. Ablations (detailed in appendix A.5) of the GNN and RNN show that both are necessary for strong performance. ECORD's contribution to the exploratory CO setting is evidencing that a suitably stochastic policy outperforms a deterministic one. To emphasise that this is not simply because of algorithmic improvements in M-DQN compared to DQN, a deterministic-acting ECORD trained with DQN is shown to also not match the performance reported in table 2.

## 5 DISCUSSION

We present ECORD, a new SOTA RL algorithm for the Max-Cut problem in term of both performance and scalability. ECORD's demonstrated efficacy on graphs with up 10 k vertices, and highly favourable computational complexity, suggests even larger problems could be tackled. By replacing multiple expensive GNN operations with a single embedding stage and rapid action-selection directed by a recurrent unit, this work highlights the importance of, and a method to achieve, efficient exploration when solving CO problems, all within the broader pursuit of scalable geometric learning.

Algorithmic improvements are a possible direction for further research. An adaptive (or learnt) temperature schedule could better trade-off stochastic exploration and deterministic solution improvement. ECORD also runs multiple optimisation trajectories in parallel, and so utilising information from other trajectories to either better inform future decisions or ensure sufficient diversity between them, would be another approach for improving performance.

With regards to further scaling improvements, any algorithm that labels vertices sequentially will at best have $\mathcal{O}(|V|)$ complexity. An interesting prospect for future work would be to note that ECORD does not rely on observing the current state of the neighbourhood of a vertex to evaluate the action quality – therefore one could envisage flipping multiple vertices in a single step, with a single centralised, or decentralised multi-agent, system.

An alternative direction would be to apply our algorithm to other complex problems and, in principle, ECORD could be applied to any vertex-labelling CO problem defined on a graph. However, its use of 'peeks' (one-step look aheads for each action), does not directly translate to problems where the quality of intermediate actions are not naturally evaluated (e.g. those where an arbitrary node labelling may be invalid such as maximum clique). Considering only valid actions or utilising indirect metrics of solution quality are possible solutions, but this remains a topic for future research.

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

# A    APPENDIX

## A.1    ARCHITECTURE DETAILS

Here we provide further details on the network architecture described in section 3.2 and figure 1 of the main text. For simplicity, we implicitly drop any bias terms in the below equations.

**Graph Neural Network**. The gated graph convolution network of Li et al. (2015) is modified to include layer normalisation, $\mathrm{LN}(\cdot)$, where the 16-dimensional embedding of node $i$ at layer $l + 1$ is given by,

$$x_i^{(l+1)} = \mathrm{LN}(\mathrm{GRU}(x_i^{(l)}, m_i^{(l+1)})), \quad m_i^{(l+1)} = \frac{1}{|N(i)|} \sum_{j \in \mathcal{N}(i)} w(e_{ij}) W_\mathrm{g} x_j^{(l)}, \qquad (4)$$

where $W_\mathrm{g} \in \mathbb{R}^{16 \times 16}$. The final embeddings after 4 rounds of message-passing are linearly projected to per-vertex embeddings, $x_i = W_\mathrm{p} x_i^{(4)}$ with $W_\mathrm{p} \in \mathbb{R}^{16 \times 16}$, that remain fixed for the remainder of the episode.

**Value network**. Recall from the main text that Q-values at time $t$ are predicted as

$$Q(v_i^{(t)}, h^{(t)}) = V(h^{(t)}) + A(v_i^{(t)}, h^{(t)}) = \mathrm{MLP_V}(h^{(t)}) + \mathrm{MLP_A}([v_i^{(t)}, W_\mathrm{h} h^{(t)}]). \qquad (5)$$

where $h^{(t)} \in \mathbb{R}^{1024}$ is the hidden state of the RNN and $v_i^{(t)} = [x_i, W_\mathrm{o} o_i^{(t)}] \in \mathbb{R}^{32}$ are the per-node embeddings with $x_i \in \mathbb{R}^{16}$, $W_\mathrm{o} \in \mathbb{R}^{16 \times 3}$ and $W_\mathrm{h} \in \mathbb{R}^{32 \times 1014}$.

The value head, $\mathrm{MLP_V}(\cdot) : \mathbb{R}^{1024} \to \mathbb{R}$, is a 2-layer network that applies a $\tanh$ activation to the input and leaky ReLU to the intermediate activations.

The advantage head, $\text{MLP}_A(\cdot) : \mathbb{R}^{64} \to \mathbb{R}$, is a 2-layer network that applies layer norm and leaky ReLU to the intermediate activations.

**Recurrent network**. The RNN hidden state is 1024-dimensional and initialised to zeros. It is updated using the embedding of the selected action, $v_*^{(t)}$, and our global observation from the next step,

$$h^{(t+1)} = \text{GRU}(h^{(t)}, m^{(t+1)}), \quad \text{where} \quad m^{(t+1)} = \text{LeakyReLU}(W_m([v_*^{(t)}, o_G^{(t+1)}])), \quad (6)$$

where $W_m \in \mathbb{R}^{64 \times 34}$.

## A.2 TRAINING DETAILS

The ECORD pseudocode is given in algorithm 1. Note that to update the network at time $t$ we make use of truncated backpropagation through time (BPTT) (Werbos, 1990; Sutskever, 2013) using the previous $t_{\text{BPTT}}$ experiences. We perform network updates every $f_{\text{upd}}$ time steps, where we update the online network parameters $\theta$ using stochastic gradient descent (SGD) on the M-DQN loss and the target network parameters $\bar{\theta}$ using a soft update (Lillicrap et al., 2015) with update parameter $\tau_{\text{upd}}$. The hyperparameters used are summarised in table 3.

Table 3: Parameters used for ECORD unless otherwise stated.

| Parameter | Value |
|---|---|
| Number of training steps | 40 000 |
| Batch size | 64 |
| Update frequency ($f_{\text{upd}}$) | 8 |
| Learning rate | 1e−3 |
| Optimizer | Adam ($\beta_1 = 0.9, \beta_2 = 0.999$) |
| Soft update rate ($\tau_{\text{upd}}$) | 0.01 |
| BPTT length ($k_{\text{BPTT}}$) | 5 |
| Buffer size ($|\mathcal{M}|$) | 40 000 |
| Discount factor ($\gamma$) | 0.7 |
| Initial exploration probability ($\varepsilon^{(t=0)}$) | 1 |
| Final exploration probability ($\varepsilon^{(t_\varepsilon)}$) | 0.05 |
| Time of exploration decay ($t_\varepsilon$) | 5000 |
| M-DQN temperature ($\tau$) | 0.01 |
| M-DQN bootstrap ($\alpha$) | 0.9 |
| M-DQN clipping ($l_0$) | -1 |

## A.3 EFFICIENT RE-CALCULATION OF CUT VALUE AND PEEKS

Given a graph, $G(V, E)$, with edge weights $w_{ij} \equiv w(e_{ij})$ for $e_{ij} \in E$, and a node labelling represented as a binary vector, $z \in \{0, 1\}^{|V|}$, the cut value is given by,

$$\begin{aligned} C(z|G) &= \frac{1}{2} \sum_{ij} w_{ij} \left( z_i(1 - z_j) + (1 - z_i)z_j \right), \\ &= \frac{1}{2} \sum_{ij} w_{ij}(z_i + z_j - 2z_i z_j). \end{aligned} \quad (7)$$

It is straightforward to decompose this into the sum of 'local' cuts

$$C(z|G) = \frac{1}{2} \sum_i C_i, \quad C_i = \sum_{j \in \mathcal{N}(i)} w_{ij}(z_i + z_j - 2z_i z_j), \quad (8)$$

where $C_i$, the cut value of the sub-graph containing node only $i$ and its neighbours, $\mathcal{N}(i)$. Similarly, we can define the total weight of un-cut edges connected to each vertex as

$$\overline{C}_i = \sum_{j \in \mathcal{N}(i)} w_{ij}(1 - z_i - z_j + 2z_i z_j). \quad (9)$$

---

**Algorithm 1:** Training ECORD

---

Initialise experience replay memory $\mathcal{M}$.
**for** each batch of episodes **do**
    Sample $B_{\mathrm{G}}$ graphs $G(V, E)$ from distribution $\mathbb{D}$
    Calculate per-vertex embeddings using GNN
    Initialise a random solution set for each graph, $S_0 \subset V$
    **for** each step $t$ in the episode **do**
        $k' = \min(t - k_{\mathrm{BPTT}}, 0)$
        **for** each graph $G_j$ in the batch **do**
            `// Flip a vertex.`
            $a^{(t)} \sim \begin{cases} \text{randomly from } V \text{ with prob. } \varepsilon \\ \mathrm{softmax}(\frac{Q_\theta(s^{(t)}, \cdot)}{\tau}) \text{ with prob. } 1 - \varepsilon \end{cases}$
            $S^{(t+1)} := \begin{cases} S^{(t)} \cup \{a^{(t)}\}, & \text{if } a^{(t)} \notin S^{(t)} \\ S^{(t)} \setminus \{a^{(t)}\}, & \text{if } a^{(t)} \in S^{(t)} \end{cases}$
            `// Add experience to buffer.`
            Add tuple $m^{(t)} = (s^{(t-k')}, \ldots, s^{(t+1)}, a^{(t-k')}, \ldots, a^{(t)}, r^{(t)}, d)$ to $\mathcal{M}$
            **if** $t \bmod f_{\mathrm{upd}} == 0$ **then**
                `// Sample batch of experiences from buffer.`
                $M^{(t)} \subset \mathcal{M}$
                `// Update online network.`
                Update $\theta$ with one SGD step using BPTT from $t$ down to $t - k'$ on $\mathcal{L}_{\mathrm{m-dqn}}$ **(??)**
                `// Update target network.`
                $\overline{\theta} \leftarrow \theta \tau_{\mathrm{upd}} + \overline{\theta}(1 - \tau_{\mathrm{upd}})$
            **end**
        **end**
    **end**
**end**
**return** $\theta$

---

The change in cut value (referred to as the 'peek' feature in the main text) if the label of vertex $i$ is flipped is then then given by

$$\begin{aligned} \Delta C_i &= \overline{C}_i - C_i, \\ &= \sum_{j \in \mathcal{N}(i)} w_{ij}(4 z_i z_j - 2(z_i + z_j) + 1), \\ &= \sum_{j \in \mathcal{N}(i)} w_{ij}(2 z_i - 1)(2 z_j - 1). \end{aligned} \quad (10)$$

Calculating these one-step look-aheads to the change in cut for each action clearly has the same complexity as calculating the cut-value itself (equation (7)). Moreover, they only have to be calculated once at the start of each episode, as when vertex $i$ is flipped from $z_i$ to $\overline{z}_i$, only the 'peeks' of vertrices $i$ and $j \in \mathcal{N}(i)$ need to be updated. These updates follow directly from the above and are given by

$$\Delta C_i \to -\Delta C_i, \quad \Delta C_j \to \Delta C_j - w_{ij}(2 z_i - 1)(2 z_j - 1). \quad (11)$$

### A.4 MCA-SOFT

MCA-soft attempts to upper bound the performance simple policies that condition actions based solely on the provided 'peeks' for each action. Denoting the known change in cut value if vertex $i$ was to be flipped as $\Delta C_i$ (see eq. (11)), MCA-soft follows a stochastic policy given by

$$a^{(t)} \sim \mathrm{softmax}\left(\frac{\Delta C_i}{\tau_{\mathrm{mca}}}\right). \quad (12)$$

To maximise the performance of MCA-soft, the temperature, $\tau_{\mathrm{mca}} \in \mathbb{R}$, is independently tuned to maximise performance on every set of graphs considered. In practice, this process is a grid search over $\tau_{\mathrm{mca}} \in \{0, 0.001, 0.003, 0.01, 0.03, 0.1, 0.3, 1, 3\}$ for results in table 1 and 4, and $\tau_{\mathrm{mca}} \in \{0, 0.0001, 0.001, 0.01, 0.1, 1\}$ for results in table 2.

## A.5 EXTENDED RESULTS

**Training on BA graphs**. For completeness, we repeat the experiments shown in the main manuscript but now training on 40-vertex BA graphs and evaluating on both ER and BA graphs with up to 500 vertices, with the results shown in Table 4.

Table 4: Scores for agents trained on BA40. Error bars denote $68\,\%$ confidence intervals. RL baseline scores are taken directly from Barrett et al. (2020) to provide the fairest possible comparison. ECORD results averaged across 3 seeds.

|  | Heuristics | | RL baselines | | |
| --- | --- | --- | --- | --- | --- |
|  | MCA | MCA-soft | S2V-DQN | ECO-DQN | ECORD |
| BA40 | $0.999^{+0.000}_{-0.006}$ | $\mathbf{1.000}^{+0.000}_{-0.000}$ | $0.961^{+0.027}_{-0.048}$ | $\mathbf{1.000}^{+0.000}_{-0.000}$ | $1.000^{+0.000}_{-0.000}$ |
| BA60 | $0.989^{+0.007}_{-0.018}$ | $0.997^{+0.008}_{-0.008}$ | $0.959^{+0.030}_{-0.040}$ | $\mathbf{1.000}^{+0.000}_{-0.000}$ | $1.000^{+0.000}_{-0.000}$ |
| BA100 | $0.965^{+0.020}_{-0.020}$ | $0.984^{+0.012}_{-0.013}$ | $0.941^{+0.037}_{-0.044}$ | $\mathbf{1.000}^{+0.000}_{-0.001}$ | $1.000^{+0.000}_{-0.001}$ |
| BA200 | $0.911^{+0.033}_{-0.037}$ | $0.929^{+0.034}_{-0.034}$ | $0.808^{+0.107}_{-0.102}$ | $\mathbf{0.983}^{+0.009}_{-0.034}$ | $0.980^{+0.020}_{-0.033}$ |
| BA500 | $0.889^{+0.015}_{-0.015}$ | $0.899^{+0.014}_{-0.014}$ | $0.499^{+0.114}_{-0.114}$ | $\mathbf{0.990}^{+0.008}_{-0.008}$ | $0.967^{+0.012}_{-0.012}$ |
| ER40 | $0.997^{+0.001}_{-0.010}$ | $\mathbf{1.000}^{+0.000}_{-0.000}$ | $0.970^{+0.020}_{-0.037}$ | $\mathbf{1.000}^{+0.000}_{-0.000}$ | $1.000^{+0.000}_{-0.000}$ |
| ER60 | $0.994^{+0.003}_{-0.013}$ | $0.999^{+0.001}_{-0.004}$ | $0.951^{+0.036}_{-0.041}$ | $\mathbf{1.000}^{+0.000}_{-0.000}$ | $1.000^{+0.000}_{-0.000}$ |
| ER100 | $0.977^{+0.017}_{-0.017}$ | $0.993^{+0.007}_{-0.008}$ | $0.941^{+0.035}_{-0.037}$ | $\mathbf{1.000}^{+0.000}_{-0.001}$ | $1.000^{+0.000}_{-0.001}$ |
| ER200 | $0.959^{+0.014}_{-0.014}$ | $0.973^{+0.012}_{-0.012}$ | $0.933^{+0.024}_{-0.024}$ | $0.994^{+0.004}_{-0.005}$ | $\mathbf{1.000}^{+0.000}_{-0.001}$ |
| ER500 | $0.941^{+0.012}_{-0.012}$ | $0.958^{+0.009}_{-0.009}$ | $0.905^{+0.019}_{-0.019}$ | $0.979^{+0.006}_{-0.006}$ | $\mathbf{0.996}^{+0.000}_{-0.004}$ |

**Baseline comparison**. Here we conduct a thorough solver comparison by training and inferring on both ER and BA graphs of up to 500 vertices. We compare ECORD to seven baselines; six taken directly from Barrett et al. (2020) (ECO-DQN, S2V-DQN, MCA, CPLEX, SimCIM, and Leleu et al. (2019); refer to Barrett et al. (2020) for implementation details), and one an extension of MCA (MCA-soft, as described in this manuscript). The results are summarised in Table 5, with the approximation ratios shown taken from averaging the solvers' performances across 100 BA and ER graphs of up to 500 vertices.

Table 5: Comparison of the approximation ratios for the Greedy (MCA and MCA-soft), branch-and-bound (CPLEX), simulated annealing (SimCIM and Leleu), and RL (S2V-DQN and ECO-DQN) Max-Cut solver baselines.

|  | MCA | MCA-soft | CPLEX | SimCIM | Leleu | S2V-DQN | ECO-DQN | ECORD |
| --- | --- | --- | --- | --- | --- | --- | --- | --- |
| ER40 | 0.997 | 1.000 | 1.000 | 1.000 | 1.000 | 0.980 | 1.000 | 1.000 |
| ER60 | 0.994 | 0.999 | 1.000 | 1.000 | 1.000 | 0.973 | 1.000 | 1.000 |
| ER100 | 0.977 | 0.993 | 0.870 | 1.000 | 1.000 | 0.961 | 1.000 | 1.000 |
| ER200 | 0.959 | 0.973 | 0.460 | 0.990 | 1.000 | 0.951 | 0.999 | 1.000 |
| ER500 | 0.941 | 0.958 | 0.160 | 0.990 | 1.000 | 0.921 | 0.985 | 0.996 |
| BA40 | 0.999 | 1.000 | 1.000 | 1.000 | 1.000 | 0.961 | 1.000 | 1.000 |
| BA60 | 0.989 | 0.997 | 1.000 | 1.000 | 1.000 | 0.959 | 1.000 | 1.000 |
| BA100 | 0.965 | 0.984 | 1.000 | 0.990 | 1.000 | 0.961 | 1.000 | 0.996 |
| BA200 | 0.911 | 0.929 | 0.830 | 0.990 | 0.940 | 0.808 | 0.983 | 0.980 |
| BA500 | 0.889 | 0.899 | 0.170 | 0.970 | 1.000 | 0.499 | 0.990 | 0.967 |

**Semidefinite programming comparison**. In addition to heuristics, another important branch of Max-Cut solver research is that of approximation algorithms. Such algorithms can offer a theoretical guarantee on the approximation ratio while still solving a relaxed formulation of the original problem in polynomial time. One such approximation method is the canonical semidefinite programming (SDP) approach of Goemans & Williamson (1995). Goemans & Williamson (1995) first formulate Max-Cut as an SDP by framing the objective as a linear function of a symmetric matrix subject to linear equality constraints (as in a linear programme) but with the additional constraint that the matrix must be positive semidefinite (whereby, for an $n \times n$ matrix $A$, $\forall x \in \mathcal{R}^n, x^t A x \geq 0$). This relaxed Max-Cut SDP formulation can be solved efficiently using algorithms such as a generalised Simplex method (Pólik & Terlaky, 2010). The insight of Goemans & Williamson (1995) was to then apply a geometric *randomised rounding* technique to convert the SDP solution into a feasible Max-Cut solution. Crucially, the randomised rounding method gives a guarantee to be within at least 0.87856 times the optimal Max-Cut value.

To the best of our knowledge, no open-access Goemans & Williamson (1995) solver exists which can handle Max-Cut problems with negatively weighted edges (Hong, 2008). However, G1-5 of the GSet graphs used in Table 2 of this manuscript have all-positive edge weights, therefore we ran the open-source cvx solver (`https://github.com/hermish/cvx-graph-algorithms`), which implements Goemans & Williamson (1995), on these 5 problems to obtain approximation ratios of $0.971, 0.970, 0.975, 0.970,$ and $0.967$ in $449, 484, 496, 521,$ and $599$ s respectively. In addition to ECORD outperforming Goemans & Williamson (1995) on these 5 GSet graphs (see Table 2) in both solving time and optimality, we note that ECORD also exceeds the $0.87856$ approximation ratio guarantee across all the GSet, ER, and BA graphs examined in our work (see Tables 1, 2, 4, and 5).

**Ablations**. We provide ablations to further investigate the key components of ECORD, highlighted in the main text as

1. a GNN to encode the spatial structure of the problem,
2. a rapid decoding conditioned on local per-node observations and an RNN's internal state that represents the optimisation trajectory,
3. an exploratory CO setting.

We ablate the GNN and RNN by using fixed zero-vectors in-place of the static per-node embeddings, $x_i$, and the RNN hidden state, $h^{(t)}$, respectively. As the advantage of exploratory CO has already been demonstrated in prior works (Barrett et al., 2020), we instead ablate the stochastic policies we find provide further improved exploration at test time (see section 4.3 of the main text). Specifically, we train the agent using DQN instead of M-DQN, as this optimises for a deterministic policy.

For all ablations, we otherwise use the same procedure as the full ECORD agent from section 4.3 of the main text, including tuning temperature of the agents policy with a grid-search. DQN is, unsurprisingly, found to be best at $\tau = 0$, as is the agent with the GNN ablated. The agent with the RNN ablated uses $\tau$ of $1e-4$, $8.5e-5$ and $5e-5$ for For G55, G60 and G70, respectively, with $\tau=5e-4$ for all other graphs. Results are presented in table 6 with ECORD significantly outperforming all ablations on larger graphs.

Table 6: ECORD, including both a GNN and RNN trained with M-DQN, and ablations (as described in the text) on the GSet graphs given a time budget of 10, 30, and 180 s (best in **bold**).

| | | No GNN | | | No RNN | | | DQN | | | ECORD | | |
|---|---|---|---|---|---|---|---|---|---|---|---|---|---|
| Graph | $\lvert V \rvert$ | 10 s | 30 s | 180 s | 10 s | 30 s | 180 s | 10 s | 30 s | 180 s | 10 s | 30 s | 180 s |
| G1 | 800 | **1.000** | — | — | 0.998 | **1.000** | — | 0.996 | 0.999 | 0.999 | 0.998 | **1.000** | — |
| G2 | 800 | 0.999 | **1.000** | — | 0.998 | 0.998 | 0.999 | 0.998 | 0.998 | 0.999 | 0.998 | 0.998 | **1.000** |
| G3 | 800 | 0.999 | **1.000** | — | 0.998 | 0.999 | 0.999 | 0.996 | 0.999 | 0.999 | 0.999 | **1.000** | — |
| G4 | 800 | **1.000** | — | — | 0.999 | **1.000** | — | 0.998 | 0.999 | 0.999 | 0.999 | **1.000** | — |
| G5 | 800 | **1.000** | — | — | 0.999 | 0.999 | 0.999 | 0.998 | 0.998 | 0.998 | **1.000** | — | — |
| G43 | 1000 | 0.997 | 0.998 | 0.999 | 0.996 | 0.999 | 0.999 | 0.991 | 0.991 | 0.991 | 0.996 | 0.997 | **1.000** |
| G44 | 1000 | 0.997 | 0.998 | **1.000** | 0.999 | 0.999 | 0.999 | 0.995 | 0.995 | 0.995 | 0.999 | 0.999 | **1.000** |
| G45 | 1000 | 0.995 | 0.997 | 0.999 | 0.997 | **1.000** | — | 0.994 | 0.994 | 0.994 | 0.998 | 0.999 | **0.999** |
| G46 | 1000 | 0.997 | 0.998 | 0.999 | 0.995 | 0.996 | 0.998 | 0.995 | 0.995 | 0.995 | 0.998 | 0.999 | **1.000** |
| G47 | 1000 | 0.996 | 0.998 | **1.000** | 0.996 | 0.996 | **1.000** | 0.992 | 0.993 | 0.993 | 0.997 | 0.998 | 0.998 |
| G22 | 2000 | 0.990 | 0.991 | 0.994 | 0.990 | 0.995 | 0.999 | 0.987 | 0.989 | 0.989 | 0.990 | 0.995 | **0.997** |
| G23 | 2000 | 0.990 | 0.995 | 0.997 | 0.989 | 0.993 | 0.998 | 0.987 | 0.991 | 0.991 | 0.991 | 0.995 | **0.997** |
| G24 | 2000 | 0.989 | 0.993 | 0.998 | 0.990 | 0.993 | 0.996 | 0.987 | 0.989 | 0.989 | 0.992 | 0.994 | **0.997** |
| G25 | 2000 | 0.991 | 0.995 | 0.996 | 0.989 | 0.994 | 0.997 | 0.988 | 0.993 | 0.993 | 0.991 | 0.996 | **0.998** |
| G26 | 2000 | 0.989 | 0.994 | 0.997 | 0.991 | 0.994 | 0.998 | 0.991 | 0.994 | 0.994 | 0.991 | 0.996 | **0.999** |
| G55 | 5000 | 0.924 | 0.949 | 0.951 | 0.950 | 0.971 | 0.981 | 0.951 | 0.953 | 0.956 | 0.947 | 0.969 | **0.985** |
| G60 | 7000 | 0.796 | 0.933 | 0.950 | 0.900 | 0.953 | 0.972 | 0.918 | 0.949 | 0.951 | 0.916 | 0.960 | **0.981** |
| G70 | 10 000 | 0.703 | 0.703 | 0.931 | 0.766 | 0.837 | 0.879 | 0.761 | 0.929 | 0.933 | 0.837 | 0.944 | **0.972** |

**Graph network timing**. As stated in the main text, the single pass of the graph neural network is negligible compared to the overall run time of ECORD. To be concrete, on the largest graph for which results are reported (G70 with $\lvert V \rvert = 10$ k nodes), our embedding stage takes $(1.96 \pm 0.09)$ ms, compared to the tens of seconds of exploratory decoding.

