# OpenReview forum: "Learning to Solve Combinatorial Problems via Efficient Exploration"
_ICLR.cc/2022/Conference — ICLR 2022 Submitted_

### Official Review · Reviewer_B7TT · 2021-10-25

**Correctness:** 2
**Technical Novelty And Significance:** 2
**Empirical Novelty And Significance:** 2
**Recommendation:** 3
**Confidence:** 4

**Main Review:**

**Strengths**
- Well written paper, it was nice and easy to read
- The paper presents a sensible approach and good motivation, where it specifically designs an architecture to minimize the computational burden of the algorithm
- The paper addresses scaling, which is an important challenge in neural combinatorial optimization
- The paper applies reasonable baselines such as the heuristic strategy (MCA/MCA-soft)
- The paper takes care to conduct fair comparison (e.g. own implementation with optimizations also for ECO-DQN baseline)

**Weakness**
- The paper only considers maximum cut, while title suggests general combinatorial optimization. As multiple aspects (flipping, peek feature) are specific to max-cut it is unclear how the proposed method can generalize to other problems. I think 'Learning to Solve Maximum-Cut' would be a more suitable title for this work.
- The approach, presented as novel RL algorithm, is very similar to ECO-DQN, but the differences are not explained, so it is unclear what are contributions and what is 'reused' from ECO-DQN. From skimming the ECO-DQN paper it seems the major differences are the architecture (GNN 'encoder' + cheap RNN policy rather than expensive GNN for every step) and algorithm (different DQN variant).
- The term SOTA is confusing, maybe even misleading. I don't think RL/GNN is clearly SOTA for maximum cut in general (see e.g. the Leleu et al. baseline in the ECO-DQN paper), which is suggested in the abstract and title of section 4.1. To avoid confusion, I think the authors should use 'best RL-based method' and keep the term SOTA for the best exact/heuristic/RL solver in general. When claiming SOTA, I think the paper should also refer to relevant max-cut literature.
- The experiments do not support the strong claims: generalization is only compared against ECO-DQN and not against heuristic baselines. Also, some general baselines are missing, especially LeLeu et al. from the ECO-DQN paper.
- The theoretical novelty is limited as the 'learning to explore' idea and max-cut setup is from ECO-DQN (Barrett et al.). The idea of encoder + (rapid) decoder architecture for combinatorial problems is similar to Drori et al. (and some earlier works on neural combinatorial optimization).

Overall, I like the paper but I think it is too incremental and too narrowly focused on max-cut to be published at ICLR.

**Detailed comments/questions/suggestions:**
- 'to equal or surpasses' -> 'to equal or surpass'
- Do MCA-soft and MCA also use 50 trajectories? (I assume so but it is not mentioned)
- Fig 2b displays linear step time, but you also run 2|V| steps so more steps for larger graphs so overall time is quadratic? Also, does blue correspond to ECORD? (labels in Fig 2a?)
- If you did a reproduction of ECO-DQN (which is a great thing!) I think it would be helpful to list both original and reproduced results for clarity in Table 1 (unless they are very similar)
- 'Heurisitcs' typo Table 1
- It may be good to mention other solution strategies are possible as well, e.g. rounding a relaxed solution (https://www.cl.cam.ac.uk/teaching/1617/AdvAlgo/maxcut.pdf)
- How does MCA and MCA-soft perform in Table 2?
- What defines exactly ECORD when claiming/suggesting that 'in principle it could be applied to any CO problem defined on a graph' (in discussion)?
- For the GNN 'encoder', is the input graph fully connected? If so, how does the 'encoder' scale?
- Given the focus on max-cut only, I think some more related work on max-cut (non-RL) could be included.

**Summary Of The Paper:**

This paper describes an RL based method for learning to solve the weighted maximum cut problem. Specifically, it proposes to process an input (fully connected?) graph using a gated graph recurrent network, and use the resulting embeddings as input for a computationally cheap RNN based policy that starts from a random cut (partition of nodes into two subsets) and subsequently 'flips' one of the nodes to move it to the other subset/other side of the cut. This policy ('encoder' + RNN) is trained using a variant of DQN to maximize the result obtained after 2*|V| flips, where V is the number of nodes in the graph. The policy makes use of a so called 'peek', a one-step lookahead of the result of each flip action on the current objective. At test time, the policy constructs multiple (deterministic) trajectories in parallel starting from different (random) initial solutions, where (I assume) the best overall result is returned. In experiments, the resulting algorithm, which is called ECORD, is shown to perform favorably when compared against ECO-DQN (Barrett et al. 2020), which, from skimming the paper, is very similar in strategy but uses a less efficient architecture and a different variant of DQN.

**Summary Of The Review:**

Overall, I like the paper, which is quite nice to read and presents a sensible and effective approach for solving the maximum cut problem using RL. Especially, the authors specifically addressed scaling to larger instances by making the architecture more efficient, which is an important and challenging topic.

Unfortunately, I still think the paper should be rejected given that
1) it is too much focused on max-cut and unclear how this framework can solve general combinatorial problems, as the title suggests
2) it is quite incremental to Barrett et al. (2020) AND not clear about the differences and
3) some important baselines are missing.

---

> ### Author Response · Authors · 2021-11-21
> **Response to Reviewer B7TT: part 1**
>
> We thank the reviewer for their time and comments which in the context of the original manuscript we accept as fair critiques.  However, we do believe that all the concerns raised can be addressed, and have provided significant additional results to do so, as well as detailed point-by-point responses below.  In our opinion, the modified submission is considerably stronger as a result, and so we are grateful to the reviewer for their help in this process.
>
> > The paper only considers maximum cut, while title suggests general combinatorial optimization...multiple aspects (flipping, peek feature) are specific to max-cut it is unclear how the proposed method can generalize to other problems...
>
> The Max-Cut problem encapsulates multiple CO problems that would commonly be considered independent (such as graph colouring, clique cover, knapsack, steiner tree) [1,2], and therefore a solver for Max-Cut can be considered to solve many important classes of CO problems.  We briefly address this in the introduction saying,
> *“Experimentally, we consider the Maximum Cut (Max-Cut) problem...chosen because of its generality (11 of the 21 NP-complete problems presented by Karp (1972) can be reduced to Max-Cut)”*,
> however we accept that this point is insufficiently explored.  *Therefore, we have amend the manuscript in the following ways*:
> - **Added explicit discussion of the CO problems covered by Max-Cut.** (2nd last para sec 1)
> - **Further underline the significance of Max-Cut by pointing to significant research and industry effort into tacking specifically Max-Cut; such as Toshiba’s simulated bifurcation machine (SBM) [3], D-Wave’s quantum annealer [4] and Coherent Ising Machines [5].** (2nd last para sec 1)
> - **For the avoidance of doubt, we have re-titled the paper “Learning to solve combinatorial graph partitioning problems via efficient exploration”.**
>
> With respect to whether ECORD is applicable to problems beyond Max-Cut and its derivatives - we believe that it is, but that the reviewer is correct in the assessment that this would require additional research.  To be concrete: we do not feel ‘flipping’, or more generally, node-(re)labelling, is a restrictive action space for CO problems — indeed labelling nodes is the general framework of ML4CO applications such as S2V-DQN which tackle multiple problem classes.  Peek features are certainly very useful for Max-Cut, and, whilst not strictly necessary to apply the general ECORD framework, can exist for any problem where a score can be assigned to different node labelling.  We explicitly address the classes of problems ECORD can naturally be applied to in the final paragraph of the discussion, where we state “However, its use of `peeks' (one-step look aheads for each action), does not directly translate to problems where the quality of intermediate actions are not as naturally evaluated (e.g. those where an arbitrary node labelling may be invalid such as maximum clique)” and briefly discuss possible solutions as directions for future work.
>
> [1] Karp. Reducibility Among Combinatorial Problems. In Complexity of Computer Computations, pp. 85 103. Springer, 1972.
>
> [2] Filar et al. Linearly-growing Reductions of Karp's 21 NP-complete Problems. arXiv 1902.10349.
>
> [3] Goto et al. Combinatorial optimization by simulat- ing adiabatic bifurcations in nonlinear hamiltonian systems. Science advances, 5(4):eaav2372, 2019.
>
> [4] Djidjev et al. Efcient combinatorial optimization using quantum annealing. arXiv preprint arXiv:1801.08653, 2018.
>
> [5] Yamamoto et al. Coherent Ising machines optical neural networks operating at the quantum limit. npj Quantum Information, 3(1):49, 2017.

---

> > ### Author Response · Authors · 2021-11-21
> > **Response to Reviewer B7TT: part 2**
> >
> > > - The approach, presented as novel RL algorithm, is very similar to ECO-DQN, but the differences are not explained, so it is unclear what are contributions and what is 'reused' from ECO-DQN...”
> > >
> > > - The theoretical novelty is limited as the 'learning to explore' idea and max-cut setup is from ECO-DQN (Barrett et al.). The idea of encoder + (rapid) decoder architecture for combinatorial problems is similar to Drori et al.
> >
> > We agree that ECORD builds on ECO-DQN (which itself extends S2V-DQN), and that it shares the same action space of flipping nodes.  However, the differences enumerated by the reviewer are both well motivated and empirically justified.  Whilst ECO-DQN introduced the notion of “exploratory CO” (reversible node-flipping instead of irreversible node labelling) as a way to mitigate sub-optimal action selection, in this work we make the conceptual connection that this opens the door to addressing scalability issues via cheap action decoding without sacrificing performance.  Indeed, we go further to demonstrate that using a stochastic policy (which is only possible because M-DQN also learns a stochastic policy, as opposed to DQN) allows us to achieve even stronger performance.
> >
> > Whilst Driori et al. also use a “GNN-encoder + rapid-decoder”, their work differs from ECORD in some key aspects.
> > - They consider CO problems with a strong ‘path-finding’ inductive bias, where the solution has a defined ordering of nodes (TSP/VRP) or is naturally considered as a path between points (minimum spanning tree/shortest path).  As such, the attention-based decoding is conditioned only on the current node, where action scores can be interpreted as the distance under some learned metric in the GNN embedding space. By contrast, Max-Cut (and related problems) do not have a natural ordering with the quality of a solution independent of the order in which actions are taken.  ECORD therefore uses an RNN to condition action selection on the entire optimisation trajectory.
> > - Driori et al. do not consider the exploratory CO setting, whereas we argue (as above) that this is where rapid-decoding architectures are especially fruitful, and demonstrate this by surpassing previous RL approaches.  Indeed, as previous actions inform where to search next, Driori et al’s work would be suboptimal for exploratory CO as it does not learn to represent this information.
> >
> > To clarify and justify the novel components of ECORD we have revised the manuscript as below.
> > - **We have added an explicit definition of the ECORD framework in the revised text** (sec 4.3): “The key components defining ECORD are: (i) the use of a single (GNN) encoding step to embed the problem structure, (ii) rapid decoding steps where per-node actions are conditioned on only local observations and an (RNN) learnt embedding of the optimisation trajectory, and (iii) an exploratory CO setting where actions can be reversed.”
> > - **We have provided the following ablations of ECORD on the GSet graphs: (i) removing the GNN, (ii) removing the RNN, (iii) using a deterministic (DQN learnt) policy**.  In all cases, these ablations fail to reach the performance of ECORD, with results summarised in the new “Ablations” section of 4.3, and detailed in Appendix A.5.
> >
> > > - The term SOTA is confusing, maybe even misleading...When claiming SOTA, I think the paper should also refer to relevant max-cut literature.”
> > > - It may be good to mention other solution strategies are possible as well, e.g. rounding a relaxed solution (https://www.cl.cam.ac.uk/teaching/1617/AdvAlgo/maxcut.pdf).
> > > - Given the focus on max-cut only, I think some more related work on max-cut (non-RL) could be included.
> >
> > We agree that ECORD is not the SOTA algorithm for Max-Cut, but that it is the SOTA RL algorithm (whilst we do not believe the reviewer interpreted this as a negative, we would note that the great majority of ML approaches to CO are not yet SOTA overall, though research such as ours is closing the gap).  We intended this to be very clear and in every reference in the text to SOTA is in this context.  We accept that the title of 4.1 in isolation is misleading.  **To address the reviewers concerns we have:**
> > - **Amend the title of Section 4.1 to explicitly refer to SOTA within the context of RL.**
> > - **Provide additional baselines using a commercial mixed integer programming solver, CPLEX, semidefinite programming (SDP), and leading SA algorithms (including Leleu et al.).  ECORD outperforms CPLEX and SPD, and is competitive with the strongest SA algorithm.  This is summarised section 4.1 with extended results provided in appendix A.5.**
> >
> > We agree that our original manuscript was overly focussed on the previous SOTA RL algorithm, ECO-DQN, and believe that our revised version provides important context.  As our work is not building on these additional methodologies, we believe it is more appropriate for them to be discussed as baselines as opposed to directly in the related work.

---

> > > ### Author Response · Authors · 2021-11-21
> > > **Response to Reviewer B7TT: part 3**
> > >
> > > > Do MCA-soft and MCA also use 50 trajectories? (I assume so but it is not mentioned)
> > >
> > > Yes they do - thank you for noting this omission **which has been rectified in the revised manuscript.**
> > >
> > > > Fig 2b displays linear step time, but you also run 2|V| steps so more steps for larger graphs so overall time is quadratic?
> > >
> > > Step time consists of action selection and updating the environment.  Action selection has constant time (up to the limits of parallelisation as detailed in Fig 2a), whereas the cost of updating the environment (i.e. calculating the new cut-value) is negligible using the efficient algorithm detailed in Appendix A.3 - which gives a linear run-time for an optimisation trajectory of length $2|V|$.  However, Fig 2b shows the step time when running the most possible trajectories in parallel given the GPU memory.  In this regime, the  parallelisation capacity of the GPU starts to break down (not every trajectory can be stepped in simultaneously without additional overhead).  This potential quadratic scaling is therefore hardware dependent and not inherent to ECORD, moreover in practice we find it sufficient to use a relatively modest fixed number of trajectories (50 in Sec 4.1 and 20 in sec 4.3) which does not fill the GPU memory.
> > >
> > > > Also, does blue correspond to ECORD? (labels in Fig 2a?)
> > >
> > > Yes, to ensure clarity **we have added this to the caption in addition to the existing legend.**
> > >
> > > > If you did a reproduction of ECO-DQN (which is a great thing!) I think it would be helpful to list both original and reproduced results for clarity in Table 1 (unless they are very similar)
> > >
> > > We do not believe it is necessary to provide multiple implementations of ECO-DQN in table 1, as our work is focussing on ECORD. Moreover, our reimplementation is significantly faster not because of any material changes to the ECO-DQN algorithm, but because we use a more efficient environment capable of simulating multiple optimisation trajectories in parallel with compiled code.  However, our implementation of both algorithms will be provided in the supporting code, to be released after the anonymous peer-review process.
> > >
> > > > “How does MCA and MCA-soft perform in Table 2?”
> > >
> > > **We have added the MCA-soft baseline to Table 2.**  For each graph, the temperature is independently tuned using a grid search that includes zero, i.e. the greedy MCA algorithm is subsumed into MSA-soft and therefore we do not feel both need presenting as MCA can not outperform MCA-soft.  With a fixed time-budget, we find that MCA-soft scales better than ECO-DQN to very large graphs, but ECORD outperforms both baselines across all graph sizes.
> > >
> > > > What defines exactly ECORD when claiming/suggesting that 'in principle it could be applied to any CO problem defined on a graph' (in discussion)?
> > >
> > > **As clarified in the revised text (“Ablations”, Sec 4.3)**, we define ECORD as the use of a single GNN-encoding step, with an RNN-decoder in an exploratory CO setting.  We discuss how this relates to previous works such as  ECO-DQN (GNN encoding-decoding at every step in an exploratory CO setting) and Driori et al (GNN-encoder and attention-decoder based on only the current state in a non-exploratory setting) in part 2 of our response.
> > >
> > > We believe that this framework could be applied to any CO problem on a graph, as the required framing of sequential vertex-level actions is very general.  However, we accept that certain problems would not be as naturally formulated in this way.  For example, CO problems where the solution has an ordering of vertices (e.g. TSP, VRP) could use multiple node labels for each time-step - however this may not be efficient.  As we have extended discussion on Max-Cut and related CO problems, **we have re-phrased this to be “ECORD could be applied to any vertex-labelling CO problem defined on a graph”**.
> > >
> > > > For the GNN 'encoder', is the input graph fully connected? If so, how does the 'encoder' scale?
> > >
> > > No - the GNN operates on the graph defined by the specific Max-Cut problem instance considered, therefore the scaling of the encoder depends on the graph topology.  We detail the scaling of typical GNN’s in Sec 4.2 (Theoretical complexity), however, as we state in Sec 4.2, in practice a “single graph network pass is typically negligible compared to the long exploratory phase”.   **To be concrete, we have added a section on GNN timings to appendix A.5**, stating: “on the largest graph for which results are reported (G70 with |V|=10k nodes), our embedding stage takes (1.96 +/- 0.09) ms compared to the tens of seconds of exploratory decoding.”

---

> > > > ### Comment · Reviewer_B7TT · 2021-11-26
> > > > **Thanks, good improvements, still I think ECORD is MAX-CUT solver**
> > > >
> > > > Thanks for the detailed response which answers most of my questions. I appreciate the updates to the paper and I'm impressed by the additional experiments conducted within a short time. I suggest to incorporate the results in the main paper, e.g. replacing table 1 by table 5/merging them.
> > > >
> > > > Still, I am sorry, I think ECORD is a MAX-CUT solver and (since MAX-CUT is an important problem indeed) I think it would make the paper actually better if it would not try to present it as more general, so it can be found by the right audience (interested in MAX-CUT). Otherwise I think the paper should perform experiments beyond MAX-CUT on other examples of graph partitioning problems. I know graph coloring but I think it would be helpful if the paper could at least hint at some more.
> > > >
> > > > As the paper has improved I encourage the authors to continue this work, but consider shifting the focus to a community interested in MAX-CUT, or add experiments to show the generality of ECORD beyond MAX-CUT. As I like the idea of 'learning to explore' and 'rapid decoding' in that case I would suggest to demonstrate these aspects on a wider range of CO problems, e.g. routing or scheduling.

---

> > > > > ### Author Response · Authors · 2021-11-26
> > > > > **Clarification of the reviewers concerns**
> > > > >
> > > > > We thank the reviewer for their response and are pleased that the additional results have been well received.  It would seem that the only remaining concern is that ECORD is demonstrated on Max-Cut, and not additional graph partitioning problems or different classes of CO problems.
> > > > >
> > > > > We believe (and the reviewer appears to agree) that Max-Cut is a highly relevant problem and would suggest that numerous works focus on tackling a specific problem class, such as Maximum Independent Set [1], Maximum Common Subgraph [2], TSP [3], VRP [4] or, indeed, Max-Cut [5] to name only a few.  We would hope that our contribution would be judged for the problems and methodologies it does address, rather than those it does not.  Whilst we appreciate that there can be a degree of subjective preference on which problems to research, we feel that a score of 3 is unwarranted for a paper that you otherwise appear to be satisfied with (and indeed like the central ideas around exploration and efficiency).
> > > > >
> > > > > We fully agree that ECORD solves Max-Cut, and equivalently all graph partitioning problems that can be represented as Max-Cut, and do not feel the paper makes any claim to the contrary (nor does it need to in order to be significant, as discussed in our previous responses).  We explicitly state that we consider Max-Cut in the abstract and introduction.  All experiments are, of course, also on Max-Cut, with the extension of ECORD to other problem classes only discussed as possible future work in the discussion. As such, if the reviewer feels unable to look more favourably on our work because it is not clear on the problems ECORD can solve, we would be more than happy to amend the title or any part of the paper the reviewer feels contribute to this.
> > > > >
> > > > >
> > > > > [1] Ahn et al., arXiv:2006.09607 (ICML 2021).
> > > > >
> > > > > [2] Bai et al., arXiv:2002.03129, (ICML 2021).
> > > > >
> > > > > [3] Fu et al., arXiv:2012.10658, (AAAI 2021).
> > > > >
> > > > > [4] Nazari et al. arXiv:1802.04240, (NeurIPS 2018).
> > > > >
> > > > > [5] Barrett et al., arXiv:1909.04063, (AAAI 2021).

---

> > > > > > ### Comment · Reviewer_B7TT · 2021-11-29
> > > > > > **Clarification**
> > > > > >
> > > > > > To clarify: I think the paper has some value in improving RL performance for Max-Cut, but I don't think this is enough to be accepted at a major conference like ICLR. For this, I think the paper should either be the first to consider Max-Cut with RL (which is not the case), present a better/practical way to solve MAX-CUT (which it does not, see Lelue et al, Table 5, which I think should be in the main paper), or present novel ideas that others are likely to build on.
> > > > > >
> > > > > > This last point is subjective and we can have a long discussion, but I don't see this: I think the paper merely changes the architecture and uses a different DQN variant, but uses the important ideas (i.e. learning to explore by flipping) from ECO-DQN (Barrett et al.). The MDP formulation and peek feature are *exactly as in Barrett et al.* which is not even mentioned! Besides that, I think the added value is relatively small (or may be even questioned, see e.g. Table 4 where ECO-DQN is better on the task is it actually trained for) and any applicability beyond MAX-CUT is highly speculative. Lastly, I think the paper should avoid 'marketing terminology' like 'SOTA RL' and 'combinatorial problems' when it considers one problem.
> > > > > >
> > > > > > I understand that there are other papers that consider only one problem, they may contain valuable contributions but it does not change my opinion about the current paper.

---

### Official Review · Reviewer_m4dC · 2021-11-01

**Correctness:** 3
**Technical Novelty And Significance:** 2
**Empirical Novelty And Significance:** 3
**Recommendation:** 5
**Confidence:** 3

**Main Review:**

Postiive points:

+ Combinational problem is very general and could have many applications.
+ ECORD is shown to be able to generalize to larger graphs.
+ ECORD is shown to be more efficient than the previous methods
+ The neural architecture is well designed with both GNN to learn vertex embeddings and value network with GRU to learn state embedding.
+ The paper is very well written.

Negative points:

- It is unclear to me how general the Max-Cut problem is. The Max-Cut is based on graph and only considers two labels. It is unclear whether ECORD can be applied to problems without graph structure or with more than two labels. In particular, when there are more than two labels, the actions space can be ill-defined since flipping will not make sense with more than two labels.
- It is unclear whether Max-Cut is hard enough to use RL. In table 1, even greedy heuristics can have very strong performance. In terms of efficiency, the greedy heuristics should be even more efficient than ECORD.
- It is unclear how ECORD improves the efficiency of ECO-DQN. The methodology section only introduces how ECORD is trained but never compares it with ECO-DQN.

**Summary Of The Paper:**

This paper presents a RL-based algorithm for solving the graph-based combinational optimization problem of Max-Cut. The key idea is to formulate the problem as an MDP, where the state is the embedings of all the vertices and the actions are the vertices to be flipped. The idea is implemented with a graph neural network to learn the embeddings and a recurrent encoding to encode the state representations. The network is trained with M-DQN. The proposed algorithm, named ECORD, is shown to be comparable to the previous state-of-the-art ECO-DQN and be significantly faster than ECO-DQN.

**Summary Of The Review:**

Overall, it remains unclear what combinational problems ECORD can address. The improvement over ECO-DQN or heuristics is not significant. It is unclear how ECORD improves ECO-DQN in terms of efficiency, which is the major contribution of this work.

---

> ### Author Response · Authors · 2021-11-21
> **Response to Reviewer m4dC: part 1**
>
> We thank the reviewer for their time and efforts considering our work.  Our revised manuscript places ECORD and Max-Cut in considerably more context - both through extended discussions and significant additional results; including ablations and baselines.  We believe these revisions address all concerns raised and have improved the quality of our paper, for which we are grateful to the reviewer.
>
> > It is unclear to me how general the Max-Cut problem is...It is unclear whether ECORD can be applied to problems without graph structure or with more than two labels...
>
> The Max-Cut problem encapsulates multiple CO problems that would commonly be considered independent (such as graph colouring, clique cover, knapsack, steiner tree) [1,2], and therefore a solver for Max-Cut can be considered to solve many important classes of CO problems.  We briefly address this in the introduction saying,
> *“Experimentally, we consider the Maximum Cut (Max-Cut) problem...chosen because of its generality (11 of the 21 NP-complete problems presented by Karp (1972) can be reduced to Max-Cut)”*,
> however we accept that this point is insufficiently explored.  **Therefore, we have amend the manuscript in the following ways:**
> - **Explicitly discuss the CO problems covered by Max-Cut.** (2nd last para Sec 1)
> - **Further underline the significance of Max-Cut by pointing to significant research and industry effort into tacking specifically Max-Cut; such as Toshiba’s simulated bifurcation machine (SBM) [3], D-Wave’s quantum annealer [4] and Coherent Ising Machines [5].** (2nd last para Sec 1)
> - **For the avoidance of doubt, we have re-titled the paper “Learning to solve combinatorial graph partitioning problems via efficient exploration”.**
>
> We would agree that ECORD is not directly applicable to problems without graph structure, however we do not feel that this is restrictive - a vast many CO problems are naturally represented on graphs [6] from travelling salesman and vehicle routing to coverage problem and influence maximization.  We do not feel that this is to the detriment of our work that it operates on graphs.  Whilst ECORD does not consider problems with more than two node labels, in principle this could be handled by simply extending the action space at each node.  Given binary labels, it is natural to define a “flipping” action, but we could equally have $n$ actions for $n$ labels or define a “cycling” action.  With that said, we believe that this setting is a matter for future research, though we note that recent works have considered how best to apply node-labelling ML algorithms in the context of CO problems with $n$>2 node labels [7], further validating the viability of such approaches.
>
> [1] Karp. Reducibility Among Combinatorial Problems. In Complexity of Computer Computations, pp. 85 103. Springer, 1972.
>
> [2] Filar et al. Linearly-growing Reductions of Karp's 21 NP-complete Problems. arXiv 1902.10349.
>
> [3] Goto et al. Combinatorial optimization by simulat- ing adiabatic bifurcations in nonlinear hamiltonian systems. Science advances, 5(4):eaav2372, 2019.
>
> [4] Djidjev et al. Efficient combinatorial optimization using quantum annealing. arXiv preprint arXiv:1801.08653, 2018.
>
> [5] Yamamoto et al. Coherent Ising machines optical neural networks operating at the quantum limit. npj Quantum Information, 3(1):49, 2017.
>
> [6] Peng et al. Graph Learning for Combinatorial Optimization: A Survey of State-of-the-Art.  arXiv: 2008.12646.
>
> [7] Gianinazzi et al. Learning Combinatorial Node Labeling Algorithms. arXiv: 2106.03594.

---

> > ### Author Response · Authors · 2021-11-21
> > **Response to Reviewer m4dC: part 2**
> >
> > > It is unclear whether Max-Cut is hard enough to use RL. In table 1, even greedy heuristics can have very strong performance. In terms of efficiency, the greedy heuristics should be even more efficient than ECORD.
> >
> > Max-Cut is an NP-Complete problem [1] (one of the most challenging classes of problems to solve), meaning no known method exists which can find an optimal solution in polynomial time. Subsequently, Max-Cut is very difficult to solve in reasonable times, and eventually becomes intractable as the scale of the instance being solved increases. ECORD is a valuable contribution to the CO literature because it shows how an RL method can scale to large NP-Complete Max-Cut instances while still finding solutions which are almost optimal.
> >
> > We accept that the comparison of ECORD to other methods (including greedy heuristics) was insufficient in the original manuscript.  **To directly address the reviewers concern, we have added the MCA-soft baseline to our experiments on very large graphs (table 2)**.  As MCA-soft is soft-greedy and tuned independently for every graph, it subsumes a simple greedy algorithm offering the same or better performance in equivalent runtime.  In practice, ECORD outperforms MCA-soft across all graph sizes.  More broadly, **we have provided additional baselines spanning the other leading approaches to Max-Cut**; a commercial mixed integer programming solver (CPLEX), semidefinite programming (SDP), and leading simulated annealing (SA) algorithms.  ECORD outperforms CPLEX and SPD, and is competitive with the strongest SA algorithm.  This is summarised section 4.1 with extended results provided in appendix A.5.  We believe that this strong performance of ECORD in comparison to leading non-RL approaches further emphasises that RL is indeed a suitable approach to the Max-Cut.
> >
> > > It is unclear how ECORD improves the efficiency of ECO-DQN. The methodology section only introduces how ECORD is trained but never compares it with ECO-DQN.
> >
> > For the avoidance of any ambiguity, **we have added an explicit definition of the ECORD framework in the revised text** (sec 4.3): *“The key components defining ECORD are: (i) the use of a single (GNN) encoding step to embed the problem structure, (ii) rapid decoding steps where per-node actions are conditioned on only local observations and an (RNN) leant embedding of the optimisation trajectory, and (iii) an exploratory CO setting where actions can be reversed.”*  By contrast, ECO-DQN uses a GNN decoding at every step, with no RNN or learnt representation of the optimisation trajectory, summarised in the Related Work as: *“ECO-DQN utilises an expensive GNN at each decision step and extends the overall number of decisions taken to be theoretically limitless, thereby restricting its scalability… ECORD remedies this by using an initial GNN embedding followed by a recurrent unit to balance the richness provided by graph networks with fast-action selection.”*.  **The improved efficiency of ECORD resulting from these changes, is analysed in section 4.2.**

---

> > > ### Comment · Reviewer_m4dC · 2021-11-23
> > > **Thanks for the response**
> > >
> > > Thank you for the detailed response. However, I am still not convinced.
> > >
> > > In table 5, it shows that ECORD is comparable to Leleu and SimCIM. It makes me confused about the advantage of ECORD. Since many of the exiting solvers can have near perfect performance in these problems, I expect more results on harder problems to demonstrate the advantage of ECORD.
> > >
> > > I also look forward to the experiments on generalizing ECORD to more than two labels. I expect the problem will become much harder with more labels.

---

> > > > ### Author Response · Authors · 2021-11-24
> > > > **Thank you for the fast response**
> > > >
> > > > Thank you taking the time to consider our response and, whilst we regret that you remain unconvinced, we would like to ask for clarity on these points.
> > > >
> > > > We hope you would agree that ECORD represents a clear improvement over previous RL algorithms for the Max-Cut problem, both in terms of raw performance and scalability (we believe these points are made clear in our manuscript but would be more than happy to provide explicit discussion is desired).   RL itself also has several desirable properties compared to classical methods, such as alleviating the necessity of pre-solving training instances (for supervised learning) or of handcrafting heuristics/hyperparameters for each problem distribution (e.g. SA such as SimCIM and Leleu et al, BLS).  Indeed, combinatorial optimisation problems (including graph partioning problems such as Max-Cut) are well motivated problem classes that are widely considered promising applications of RL and have accordingly attracted considerable research interest.
> > > >
> > > > We agree that ECORD's performance (Table 5) is only comparable SOTA simulated annealing heuristics (though we note that ECORD trains only on the smallest graphs and generalises to larger ones at test time).  However, we would point out that there are many fields where RL is yet to demonstrate clear superiority over all other approaches, but that this does not devalue progress in these areas (especially when, for the aforementioned reasons, RL is a highly promising methodology).  Therefore, it doesn't seem that this alone is grounds for the disqualification of our work.
> > > >
> > > > With regards to experiments on multiple labels, this is a different problem class we consider beyond the scope of our work.  Whilst certainly it represents a highly challenging task, Max-Cut is already NP-complete and can not be considered an easy problem to solve.  Besides from the theoretical complexity of the problem, the difficulty of the problems considered by ECORD compares very favourably to prior work (both RL and non-RL).  To summarise the extended details we provide in our response to reviewer 3ifn, ECORD is scaled to larger graph sizes (10k nodes) than was considered in corresponding publications of every baseline in Table 5, including SimCIM (800 nodes) and Leleu et al. (500 nodes).
> > > >
> > > > Please let us know should you require further clarification so that we can better address your concerns.  We are very grateful for your continued consideration and hope that this discussion allows you to re-visit your recommendation.

---

> > > > > ### Comment · Reviewer_m4dC · 2021-11-29
> > > > > **Thanks for your reponse**
> > > > >
> > > > > I thank the authors for the response. I agree that ECORD has clear improvement over previous RL-based methods in terms of scalability (but for raw performance, I am not convinced since the improvement is marginal to ECO-DQN).
> > > > >
> > > > > Since the main contribution is scalability, it is natural to demonstrate the advantage on harder problems, where heuristics may have poor performance (or other solvers can not scale). However, I can not be convinced in this aspect from the current experiments.
> > > > >
> > > > > I don't agree that it is ok to not demonstrate superiority of RL over other methods. The main motivation of applying RL is to improve the existing algorithms with learning. Given sophisticated neural networks and long training time of RL, if we can just be comparable to the baselines (in terms of performance and scalability), then I don't see the necessity of applying RL.
> > > > >
> > > > > I will keep my score unchanged. I think RL is promising. However, the experiments are not convinced enough. I encourage the authors to conduct more experiments on harder problems.

---

### Official Review · Reviewer_5Z4N · 2021-11-02

**Correctness:** 4
**Technical Novelty And Significance:** 3
**Empirical Novelty And Significance:** 3
**Recommendation:** 6
**Confidence:** 3

**Main Review:**

Pros:
1. The proposed approach is solid. I think this algorithm can easily extended for solving combinatorial optimization problems other than the maximum cut problem.
2. The empirical comparison is thorough with respect to the considered baselines.

Cons:
1. I am mainly concerned with the empirical comparison.
- The maximum cut problem is a representative application of semi definite programming solvers. See [1] for an example. I think this baseline is necessary for practitioners to see whether if the proposed algorithm has any useful-ness in real-world applications.
- Gset (considered in this paper) is a popular benchmark for the maximum cut problem and there are several non-DNN-based results that can easily compared with the proposed solvers. For example, the authors can compare with [2].
- To my knowledge, [3] is a relatively new GNN-based combinatorial solver that can solve maxcut. Their algorithm can be applied to maximum cut with a minor changes (they solve graph partitioning problem). This comparison is important since [2] is also a GNN-based solver with running time linear with respect to the problem size. They use a single GNN-pass and decode the solution.
2. It seems that this paper is very similar to [4] in a sense that they both use a GNN to extract node embeddings for the given problems and applies DQN-based training for a node-wise predictor to generate solutions. I think the authors can provide a more thorough comparison between two methods. My current understanding is that [4] pretrains a GNN using supervised learning and ECORD use end-to-end truncated backpropagation.

Minor comments:
1. Ablation studies on using DQN over M-DQN would be useful to see whether if the empirical improvement comes from the proposed scheme, or simply using a better RL algorithm.
2. I hope the authors could provide more reference and related works on ways to solve the maximum cut problem without using DNN.

[1] Improved approximation algorithms for maximum cut and satisfiability problems using semidefinite programming, 1995
[2] Breakout Local Search for the Max-Cut Problem, 2012
[3] Erdos Goes Neural: an Unsupervised Learning ˝ Framework for Combinatorial Optimization on Graphs, 2020
[4] Learning heuristics over large graphs via deep reinforcement learning, 2019

**Summary Of The Paper:**

This paper proposes a scalable deep neural network (DNN)-based solver for the maximum cut problem. The main idea is to use (1) a single graph neural network (GNN) pass to acquire node embeddings and (2) use sequential decoding (without using GNN) to acquire the maximum cut solution. The experiments demonstrate good performance of the proposed algorithm, in particular for speeding up the generation of solutions.

**Summary Of The Review:**

I think this paper is based on a solid idea and shows good empirical performance. However, at the current state, it is hard to access the significance of this paper since baseline algorithms used in real-world is missing. Furthermore, I would like to see description on the proposed algorithm's difference to existing work of Manchanda et al., (2019) to further support novelty of the proposed method.

---

> ### Author Response · Authors · 2021-11-21
> **Response to Reviewer 5Z4N: part 1**
>
> We are grateful to the reviewer for their time and thoughtful feedback.  In response to the overall comments regarding our empirical evaluation we have provided extensive additional results and baselines that we believe both address the reviewers concerns and considerably strengthen the paper.  We have provided detailed responses to all points raised below.
>
> > I am mainly concerned with the empirical comparison...
>
> Our choice of baselines was motivated by (i) comparing to the previous SOTA in RL (ECO-DQN) and (ii) comparing to other heuristics that frame the optimization problem as a series of per-node ‘flipping’ actions (MCA and MCA-soft).  However, we accept that we should provide broader context to our results, and so have revised the paper accordingly.
> - **We provide additional baselines spanning the other leading approaches to MAX-CUT**; a commercial mixed integer programming solver (CPLEX), semidefinite programming (SDP), and leading SA algorithms (including Leleu et al.).  ECORD outperforms CPLEX and SPD, and is competitive with the strongest SA algorithm.  This is summarised section 4.1 with extended results provided in appendix A.5.
> - **We justify our choice of baselines (as above) in the methods section of 4.1, whilst emphasising that these other approaches exist and summarising their performance compared to ECORD.**
>
> With regards to the specific works raised by the reviewer:
>
> [1]  We have included SDP as a baseline as above.  The canonical algorithm of Goemans & Williamson referenced cannot be directly applied to the ER40/BA40 to ER500/BA500 graphs we use for the rest of our extended baselines as is does not readily handle negative edge weights (if you apply it anyway, performance is very poor).  Therefore we compare to SDP on larger GSet graphs (with positive edge weights), and find that **ECORD provides significantly better solutions in reduced running time**.  In any case, we note that in all of our empirical results we exceed the 0.87856 optimality guarantee of [1].  **All of this is detailed in our extended appendix A.5**.
>
> [2] is where many of the best-bounds for the GSet come from, therefore it is implicitly used as a baseline and explicitly referenced when introducing the GSet (section 4, “Datasets”).  We feel that direct comparison would not be overly fruitful as speed would be largely determined by the engineering efforts applied, however we believe that our extended discussion of non-RL heuristics (detailed above) places ECORD in sufficient context.
>
> [3] is an interesting approach using unsupervised learning for solving graph partitioning problems.  However, the work omits comparisons to much of the SOTA ML graph partitioning literature of which we are aware, and focuses on a different subset of problems (max clique and min-cut).  Despite this, we note that in the case of graph partitioning, [3] focuses on graphs of 26, 132, and 7,512 vertices  (SF-295, Twitter, and Facebook respectively), taking 289s to solve the largest. As shown in our manuscript, ECORD scales beyond these graph sizes and solves graphs of an equivalent size in less time (see Tables 1, 2, 4, and 5), albeit on a different problem class (Max-Cut).  Additionally, [3] is neither an RL or general SOTA graph partitioning method (as stated by the authors, their *“evaluation does not intend to establish SOTA results”*).  We therefore believe that the numerous baselines we now provide, thanks to the reviewer’s useful suggestions, sufficiently benchmark ECORD relative to SOTA RL methods and other classes of solvers.
>
> > It seems that this paper is very similar to [4]...
>
> We agree that [4] is an important work with regards to scaling GNN-based methods to certain types of large CO problems. However, this scalability was achieved by pruning nodes which were considered to be unlikely to exist in the solution set, thereby significantly reducing the size of the problem to be solved.  Unfortunately, such an approach is not applicable to CO problems whose nodes cannot be pruned, e.g. those where all nodes must be correctly labelled to find the optimal solution such as Max-Cut.  Therefore, [4] cannot be used as a baseline in our work.  This is summarised in the final subsection of “Advances in scalability” in the Related Work: *“Manchanda et al. (2020) furthered the work... which precludes it from solving some of the most fundamental CO problems such as Max-Cut.”*.

---

> > ### Author Response · Authors · 2021-11-21
> > **Response to Reviewer 5Z4N: part 2**
> >
> > > Ablation studies on using DQN over M-DQN would be useful to see whether if the empirical improvement comes from the proposed scheme, or simply using a better RL algorithm.
> >
> > Thank you for the insightful suggestion.  **The ablations we have added in the revised manuscript (which are discussed in section 4.1 and detailed in appendix A.5) includes ECORD trained using DQN**.  We find that performance is significantly worse, which is unsurprising as DQN trains a deterministic policy, whereas M-DQN trains a stochastic policy (the latter is shown to be important for maximising ECORD’s performance in section 4.3).  We did attempt to train a model with DQN and then still use a soft-DQN policy at test time, however this provided worse performance than the learnt deterministic policy.  It is possible that a stochastic-acting ECO-DQN trained with M-DQN would outperform the deterministic version, however we did not investigate this as our focus is on efficient/scalable exploration, which ECO-DQN cannot provide as discussed in the main text.
> >
> > > I hope the authors could provide more reference and related works on ways to solve the maximum cut problem without using DNN.
> >
> > We agree that our original paper did not provide sufficient context with regards to other approaches to Max-Cut.  We believe that the changes we have made to extend this discussion and provide numerous additional baselines (discussed above and not repeated here for brevity) address this issue, and that the paper is better for it.

---

> > > ### Comment · Reviewer_5Z4N · 2021-11-26
> > > **Re: Response to Reviewer 5Z4N**
> > >
> > > Thank you for the detailed response. I think the authors' response was excellent, and it seems that some of my suggestions were not appropriate (i.e., comparing with [3]). **However, I hope the authors would state explicitly on how the performance of ECORD compares with [2] to avoid any misconceptions that DNN solvers are now SOTA for MAX-CUT.**
> > >
> > > My impression from the revised paper is that ECORD does need further development to become a real-world SOTA (i.e., SOTA solvers not using deep neural networks). However, I like the idea of the paper and I think that beating the real-world SOTA using DNNs can be more of a long term goal. I have raise my score based on this opinion.

---

> > > > ### Author Response · Authors · 2021-11-28
> > > > **Re: Response to Reviewer 5Z4N**
> > > >
> > > > Thank you for your kind remarks and for raising your score for our work. We agree that we should explicitly state how ECORD compares with [2] to highlight that our approach is not SOTA for Max-Cut when considering non-DNN methods, and will be sure to make these changes in the camera-ready copy of our paper.

---

### Official Review · Reviewer_3ifn · 2021-11-02

**Correctness:** 2
**Technical Novelty And Significance:** 2
**Empirical Novelty And Significance:** 2
**Recommendation:** 5
**Confidence:** 3

**Main Review:**

Although this paper has excellent motivation for improving the efficiency of current GNN based RL for CO problems, I have some concerns about the approach and experiments.
(1)	Although Maximum Cut problem is one of classic CO problem, it cannot represent all CO problem. For this reason, the title is much larger than the content of this paper, unless the authors can show the generalization of their method for other CO problem.
(2)	Restricting the GNN in the preprocessing step for embedding is a good idea for reducing the complexity in running time. According to experimental results, the paper states the performance is better than ECO-DQN. Then, it is very natural to raise the question: whether is the contribution of GNN very little for the performance? To answer this question, the authors should give some evidence of comparison with GNN-free graph embedding methods.
(3)	In the experiments, the paper only compared with ECO-DQN and two simple heuristics, which cannot support to claim that the proposed method is better than current SOTA of approximate algorithms. Some of them are efficient for large-scaled problem too.
(4)	If a max-cut problem has multiple best solutions, I wonder whether the convergence of proposed approach can be guarantee with limited learning steps? Are there some test cases for this point in experiments?
(5)	The terminating condition will affect the convergence and performance of the given approach largely. The authors used 2|V| for small-scaled instances and 4|V| for large-scaled ones. Could the authors give some theoretical analysis for the necessary number of learning steps. If could not, the algorithm is hard to be used in practice.
(6)	The largest problem targeted in experiments is about just 10000 nodes. In the era of big data, the scale for testing the scalability is not enough. Many real problem graphs have millions of nodes.


**Summary Of The Paper:**

To deal with the efficiency and scalability issues in current RL approaches for Max-Cut problem, this paper proposes a new method called ECORD, based on Munchausen DQN’s exploration with GNN pre-computed embedding of each vertex. Running Time Complexity is analyzed. Compared with current baselines, time complexity is reduced from O(|V|^3) to O(|V|^2). Experiments, are conducted on published dataset, show its superiority in efficiency and performance.

**Summary Of The Review:**

Although this paper has excellent motivation for improving the efficiency of current GNN based RL for CO problems, I have some concerns about the approach and experiments.

---

> ### Author Response · Authors · 2021-11-21
> **Response to Reviewer 3ifn: part 1**
>
> We thank the reviewer for their feedback and have provided significant additional results to address their concerns, as detailed in our point-by-point responses below.  We are grateful for the reviewers guidance in improving our manuscript, and would be happy to address any additional points in forthcoming discussions.
>
> > (1) Although Maximum Cut problem is one of classic CO problem, it cannot represent all CO problem. For this reason, the title is much larger than the content of this paper, unless the authors can show the generalization of their method for other CO problem.
>
> The Max-Cut problem encapsulates multiple CO problems that would commonly be considered independent (such as graph colouring, clique cover, knapsack, steiner tree) [1,2], and therefore a solver for Max-Cut can be considered to solve many important classes of CO problems.  We briefly address this in the introduction saying, *“we consider the Maximum Cut (Max-Cut) problem ... because of its generality (11 of the 21 NP-complete problems presented by Karp (1972) can be reduced to Max-Cut)”*, however we accept that this point is insufficiently explored.  **Therefore, we have amend the manuscript in the following ways:**
> - **Explicitly discuss the CO problems covered by Max-Cut.** (2nd last para. in Sec 1.)
> - **Further underline the significance of Max-Cut by pointing to significant research and industry effort into tacking specifically Max-Cut; such as Toshiba’s simulated bifurcation machine (SBM) [3], D-Wave’s quantum annealer [4] and Coherent Ising Machines [5].**  (2nd last para. in Sec 1.)
> - **For the avoidance of doubt, we re-titled the paper “Learning to solve combinatorial graph partitioning problems via efficient exploration”.**
>
> [1] Karp. Reducibility Among Combinatorial Problems. In Complexity of Computer Computations, pp. 85 103. Springer, 1972.
>
> [2] Filar et al. Linearly-growing Reductions of Karp's 21 NP-complete Problems. arXiv 1902.10349.
>
> [3] Goto et al. Combinatorial optimization by simulating adiabatic bifurcations in nonlinear hamiltonian systems. Science advances, 5(4):eaav2372, 2019.
>
> [4] Djidjev et al. Effcient combinatorial optimization using quantum annealing.  arXiv:1801.08653.
>
> [5] Yamamoto et al. Coherent Ising machines optical neural networks operating at the quantum limit. npj Quantum Information, 3(1):49, 2017.
>
> > (2) Restricting the GNN in the preprocessing step for embedding is a good idea for reducing the complexity in running time. According to experimental results, the paper states the performance is better than ECO-DQN. Then, it is very natural to raise the question: whether is the contribution of GNN very little for the performance? To answer this question, the authors should give some evidence of comparison with GNN-free graph embedding methods.
>
> We agree that this is a natural question.  **In the revised manuscript we have provided ablations on the large (GSet) graphs, summarised in Sec 4.3 and detailed in Appendix A.5, including removing the GNN**.  These results show that the GNN does indeed significantly contribute to performance.  Whilst it may be possible to capture the necessary information about the structure of the CO problem using neural-network free graph embeddings (for example, using handcrafted node-features or Laplacian position eigenvectors [6]), we consider this beyond the scope of our current work.  Besides, we have also provided the timings of the GNN in the extended results (Appendix A.5) to underline that this is negligible in the overall running time of ECORD.
>
> [6] Dwivedi et al. A Generalization of Transformer Networks to Graph. arXiv 2012.09699.
>
> > (3) In the experiments, the paper only compared with ECO-DQN and two simple heuristics, which cannot support to claim that the proposed method is better than current SOTA of approximate algorithms. Some of them are efficient for large-scaled problem too.
>
> We only claim ECORD is SOTA in the context of RL algorithms (although reviewer B7TT noted that the title of section 4.1 in isolation is misleading, **which we have now corrected**).  Our choice of baselines was motivated by (i) comparing to the previous SOTA in RL (ECO-DQN) and (ii) comparing to other heuristics that frame the optimization problem as a series of per-node ‘flipping’ actions (MCA and MCA-soft).  However, we accept that we should provide broader context to our results, and so have revised the paper accordingly:
>
> - **We provide additional baselines spanning the other leading approaches to MAX-CUT; a commercial mixed integer programming solver (CPLEX), semidefinite programming (SDP), and leading simulated annealing algorithms.**  ECORD outperforms CPLEX and SPD, and is competitive with the strongest SA algorithm.  This is summarised in sec. 4.1 with extended results provided in appendix.
> - **We justify our choice of baselines (as above) in the methods sec. of 4.1, whilst emphasising that these other approaches exist and summarising their performance compared to ECORD.**

---

> > ### Author Response · Authors · 2021-11-21
> > **Response to Reviewer 3ifn: part 2**
> >
> > > (4) If a max-cut problem has multiple best solutions, I wonder whether the convergence of proposed approach can be guarantee with limited learning steps? Are there some test cases for this point in experiments?
> >
> > Max-Cut is NP-complete, therefore we can’t guarantee finding  an optimal solution within a favourable (polynomial) running time - instead heuristics seek to provide strong practical performance without such guarantees.  Strictly, as ECORD learns a stochastic policy and can always access any configuration in at most $|V|$ actions (for an $|V|$-vertex graph), it’s ergodic exploration does guarantee that the optimal solution will eventually be found, however this could also be claimed for simple random sampling and therefore we do not feel explicit discussion would benefit the paper.  **We believe the changes discussed in our responses to point (3) - where ECORD is demonstrated to outperform an SDP algorithm that does provide theoretical guarantees of solution quality - and our discussion in response to point (5) - clarifying possible bounds on the time to solve an instance - will also help address this question.**
> >
> > > (5) The terminating condition will affect the convergence and performance of the given approach largely. The authors used 2|V| for small-scaled instances and 4|V| for large-scaled ones. Could the authors give some theoretical analysis for the necessary number of learning steps. If could not, the algorithm is hard to be used in practice.
> >
> > As Max-Cut is NP-complete, it is not possible to provide guarantees of finding the solution in non-polynomial time.  Since any guarantees could not be tighter than the time taken to solve by brute force, we do not feel it is practically useful to try and provide such theoretical analysis.  A lower-bound on the required solving time is $|V|/2$ actions (i.e. the maximum number of actions required to reach an arbitrary configuration, up to a global flip of all vertices, from a random initial state).
> >
> > As it is not possible to provide meaningful bounds for our heuristic methods, we instead evaluate performance using the most practical metric of wall-clock time on larger graphs.  We only use $2|V|$ steps only on smaller graphs (sec 4.1), for consistency with previous works using this dataset and $4|V|$ steps only for our investigation of stochasticity (Fig 3).  On large graphs, we demonstrate performance significantly beyond previous RL methods in 3 minutes.

---

> > > ### Author Response · Authors · 2021-11-21
> > > **Response to Reviewer 3ifn: part 3**
> > >
> > > > (6) The largest problem targeted in experiments is about just 10000 nodes. In the era of big data, the scale for testing the scalability is not enough. Many real problem graphs have millions of nodes.
> > >
> > > We fully agree that scalability is vital to real-world applicability and, indeed, this is one of the problems that motivates ECORD.  However, the size of a graph must be taken in the context of the problem being solved over it - whilst many problems have millions of nodes - Max-Cut is especially challenging to scale as every node must be correctly labelled to find the solution (as opposed to, for example, recent ML4CO work on coverage problems [7], which can be scaled to much larger graphs as only a small sub-section of the nodes contribute to the solution.)  Our results use the GSet, a canonical and long-studied dataset for Max-Cut, and 10k is the largest graph available for us to test on [8,9] - therefore it would not be straightforward to find larger instances for which optimal (or, at least, very good) solutions are known.
> > >
> > > Indeed, ECORD compares very favourably to prior work (both RL and non-RL) in terms of scalability.
> > > - Amongst the relevant literature of which we are aware, the largest graph tackled by any previous RL method for Max-Cut is [10], who infer on graphs of up to 2,000 vertices (G22-32 from GSet). Other RL works such as Beloborodov et al. [11] and Driori et al. [12] scale to 800 and 1,000 vertices respectively. By contrast, ECORD is able to outperform our baselines on up to 10,000-vertex graphs in only 180 seconds (G70 from GSet, see Table 2).
> > > - Of the relevant non-RL ML-based approaches, Leleu et al. [13] (referenced by reviewer B7TT and included in our revised manuscript) infer on graphs with up to 200 vertices, though it has since been demonstrated up to 500 vertices in [2].  The Breakout Local Search algorithm proposed by Benlic and Hao [14], referred to by reviewer 5Z4N and referenced in our paper, is a very strong Max-Cut heuristic, scaling to 20,000 vertices, albeit with 5.6 hours runtime, and therefore establishing a SOTA Max-Cut solver which RL is yet to beat.
> > > - In our updated manuscript, we have also added a new comparison to the SDP approximation method of Goemans and Williamson [15] (see appendix A.5), and we show ECORD outperforming SDP on G1-5 (the 800-vertex GSet graphs with all-positive edge weights) both in terms of optimality and solving time.
> > >
> > > **We accept that this context is not sufficiently clear in the original submission and so, in addition to our pre-existing comment that Max-Cut “presents a challenging problem for scalable CO as every vertex must be correctly labelled in the final solution, meaning there is no reasonable way to restrict our optimisation to only a subset of vertices.”, we have extended our discussion or other works on Max-Cut, both explicitly and with additional baselines (see our response to point (3)).**
> > >
> > > [7] Manchanda et al. Learning heuristics over large graphs via deep reinforcement learning. arXiv:1903.03332.
> > >
> > > [8] https://sparse.tamu.edu/Gset
> > >
> > > [9] https://www.cise.ufl.edu/research/sparse/mat/Gset/README.txt
> > >
> > > [10] Barrett et al. Exploratory combinatorial optimization with reinforcement learning. In Proceedings of the AAAI Conference on Articial Intelligence, volume 34, pp. 3243 3250, 2020.
> > >
> > > [11] Beloborodov et al. Reinforcement Learning Enhanced Quantum-inspired Algorithm for Combinatorial Optimization. Machine Learning: Science and Technology 2 025009, 2020.
> > >
> > > [12] Drori et al. Learning to Solve Combinatorial Optimization Problems on Real-World Graphs in Linear Time. arXiv:2006.03750, 2020.
> > >
> > > [13] Timothe Leleu et al. Destabilization of local minima in analog spin systems by correction of amplitude heterogeneity. Phys. Rev. Lett., 122:040607, 2019.
> > >
> > > [14] Una Benlic et al. Breakout Local Search for the Max-Cut problem. Engineering Applications of Artificial Intelligence, 26(3):1162 1173, 2013. ISSN 0952-1976.
> > >
> > > [15] Michel X. Goemans et al. Improved Approximation Algorithms for Maximum Cut and Satisfiability Problems Using Semidefinite Programming. Association for Computing Machinery, ISSN 0004-5411, 1995.

---

### Author Response · Authors · 2021-11-28
**Final request for clarification of reviewers concerns**

We are grateful to the reviewers for their feedback, however with the discussion deadline approaching, we would like to ask again for clarification of the reviewers concerns, as these do not seem to align with the scores given.  After our revisions, there does not appear to be any outstanding questions regarding our methodology or experimental results.

- Reviewer m4dC appears unable to endorse the paper because (i) our algorithm is only comparable in performance to non-ML baselines (specifically SOTA simulated annealing heuristics) and (ii) would like to see results on "harder" problems than Max-Cut.  As argued in our response below, all reviewers appear to agree that our algorithm significantly advances the SOTA in ML approaches to Max-Cut, and Max-Cut itself is NP-Complete with significant research and commercial efforts made to tackle precisely the problem setting we consider.  We do not believe that ML research must significantly outperform all other methodologies before it can be considered worthwhile if, as is the case for ECORD, it makes significant advancements in both performance and scalability.  Moreover, to argue that Max-Cut is not sufficiently hard or applicable to be of interest is, in our opinion, simply incorrect.  We appreciate the reviewers opinions on these points may differ, but would like them to be explicit if this is why they will not raise their rating from a 5, as we would hope these are minority views.  Of course, if there are other concerns that have not been discussed, we would be grateful for the chance to address them before the discussion period ends.

- Reviewer B7TT's only remaining criticism also appears to be that "ECORD is a Max-Cut solver", a stance we agree with but do not understand why it is a negative.  We don't believe it can be argued that Max-Cut is not sufficiently hard or interesting (as discussed above and in detail in our responses).  Alternatively, if the reviewer feels the paper is not clear on the problem class tacked by ECORD, we remain very open to clarifying any part of the manuscript that may contribute to this impression.  However, at current, we are confused as to why the problem class we consider justifies a score of 3.

We believe the paper has been significantly improved as a result of integrating the questions of all reviewers and it is in this spirit that we are seeking clarification on the above.

---

### Decision · Program_Chairs · 2022-01-20

**Decision:**

Reject

**Comment:**

The paper proposes an efficient RL-based approach for solving the weighted maximum cut problem. The proposed approach shares high-level insights with prior work such as ECO-DQN (Barrett et al.) and S2V-DQN; the key contribution is to demonstrate that the proposed cheap action decoding and stochastic policy strategy can improve the scalability without sacrificing much of the quality of the solution on the tasks considered in this paper.

The reviewers in general find the paper well presented, and especially note that the clear motivation for improving the efficiency of current GNN-based RL baselines, particularly represented by ECO-DQN.

A common concern among the reviewers is that the original title is misleading; the authors acknowledge that they should properly position the paper to avoid confusion that they were to address general combinatorial optimization problems (as the current title suggests). Notably, many combinational optimization problems can be reduced to max-cut as suggested in the authors’ responses; demonstrating the performance in (some of) these problems via a max-cut reduction would be helpful to support the significance of this work.

Beyond the title and positioning of this work, there were also initial confusions among the committee in terms of the choice of both (RL or supervised) learning-based and heuristic-based baselines. The authors did an excellent job in clarifying many of the questions in terms of related work and baselines (the clarity of the work has improved over the rebuttal phase). However, despite the additional ablation study and newly added baselines, there remain concerns/questions in the choice of task domains (lack of hard problem instances where existing solvers, learning- or heuristics -based may fail due to (possibly higher) computational complexity). Given the empirical focus of the paper, this appears to be an important concern, and not all reviewers are convinced the current empirical results are significant to warrant acceptance of this work.